# Upregulation of TRPM3 in nociceptors innervating inflamed tissue

**Marie Mulier[1,2], Nele Van Ranst[1,2], Nikky Corthout[3,4], Sebastian Munck[3,4], Pieter Vanden Berghe[5], Joris Vriens[6], Thomas Voets[1,2†]\*, Lauri Moilanen[1,2†‡]**

[1]Laboratory of Ion Channel Research (LICR), VIB-KU Leuven Centre for Brain & Disease Research, Leuven, Belgium; [2]Department of Cellular and Molecular Medicine, KU Leuven, Leuven, Belgium; [3]VIB Bio Imaging Core and VIB-KU Leuven Centre for Brain & Disease Research, Leuven, Belgium; [4]Department of Neuroscience, KU Leuven, Leuven, Belgium; [5]Laboratory for Enteric NeuroScience (LENS), TARGID, Department of Chronic Diseases Metabolism and Ageing, KU Leuven, Leuven, Belgium; [6]Laboratory of Endometrium, Endometriosis and Reproductive Medicine, G-PURE, Department of Development and Regeneration, KU Leuven, Leuven, Belgium

**\*For correspondence:**
thomas.voets@kuleuven.vib.be

†These authors contributed equally to this work

**Present address:** ‡Department of Pathology, Central Finland Health Care District, Jyväskylä, Finland

**Competing interests:** The authors declare that no competing interests exist.

**Abstract** Genetic ablation or pharmacological inhibition of the heat-activated cation channel TRPM3 alleviates inflammatory heat hyperalgesia, but the underlying mechanisms are unknown. We induced unilateral inflammation of the hind paw in mice, and directly compared expression and function of TRPM3 and two other heat-activated TRP channels (TRPV1 and TRPA1) in sensory neurons innervating the ipsilateral and contralateral paw. We detected increased *Trpm3* mRNA levels in dorsal root ganglion neurons innervating the inflamed paw, and augmented TRP channel-mediated calcium responses, both in the cell bodies and the intact peripheral endings of nociceptors. In particular, inflammation provoked a pronounced increase in nociceptors with functional co-expression of TRPM3, TRPV1 and TRPA1. Finally, pharmacological inhibition of TRPM3 dampened TRPV1- and TRPA1-mediated responses in nociceptors innervating the inflamed paw, but not in those innervating healthy tissue. These insights into the mechanisms underlying inflammatory heat hypersensitivity provide a rationale for developing TRPM3 antagonists to treat pathological pain.

## Introduction

Painful stimuli are detected by nociceptors in peripheral tissues and are transmitted as action potentials towards the central nervous system to elicit pain sensation (*Viana et al., 2019*; *von Hehn et al., 2012*; *Vriens et al., 2014*). Under pathological conditions, such as inflammation or tissue injury, nociceptors become sensitized to mechanical and thermal stimuli. Such nociceptor hypersensitivity gives rise to allodynia (the sensation of pain to a stimulus that is usually not painful), hyperalgesia (increased pain sensation to a stimulus that is usually painful) or to spontaneous pain without any clear stimulus (*Viana et al., 2019*; *von Hehn et al., 2012*; *Jensen and Finnerup, 2014*). The current analgesic drug therapies, including non-steroidal anti-inflammatory drugs, opioids, gabapentinoids, and antidepressants, are often limited in efficacy in a large number of pain patients and may have severe adverse effects that limit their use (*Colloca et al., 2017*; *Skolnick, 2018*; *Voets et al., 2019*). The need for better and safer analgesic drugs is painfully illustrated by the dramatic rise in opioid addiction and related deaths, known as the opioid crisis (*Skolnick, 2018*). The search for new analgesic drugs with novel mechanisms of action inevitably depends on a deep understanding of the cellular and molecular mechanisms underlying nociceptor sensitization (*Colloca et al., 2017*).

In this context, several members of the transient receptor potential (TRP) superfamily of cation channels play key roles as primary molecular sensors in nociceptor neurons, directly involved in translating external stimuli into neuronal activity and pain (*Basbaum et al., 2009*). For instance, we recently demonstrated that heat-induced pain in mice depends on a trio of heat-activated TRP channels, TRPM3, TRPA1, and TRPV1 (*Vandewauw et al., 2018*). Robust neuronal and behavioral heat responses were observed as long as at least one of these three TRP channels was functional, but were fully abolished in $Trpm3^{-/-}/Trpv1^{-/-}/Trpa1^{-/-}$ triple knockout mice (*Vandewauw et al., 2018*). These results indicate triple redundancy of the molecular sensors for acute heat. Intriguingly, however, there apparently is no such redundancy for the development of heat hypersensitivity in inflammatory conditions. Indeed, genetic ablation or pharmacological inhibition of either only TRPV1 (*Caterina et al., 2000*; *Davis et al., 2000*; *Gavva et al., 2005*; *Honore et al., 2005*) or only TRPM3 (*Alkhatib et al., 2019*; *Vriens et al., 2011*; *Straub et al., 2013*) is sufficient to fully suppress inflammatory heat hyperalgesia. However, the precise mechanisms and the relative contributions of the heat-activated TRP channels to nociceptor sensitization under inflammatory conditions remain incompletely understood.

To address this problem, we induced experimental inflammation in one hindpaw of mice and evaluated changes in expression and function of TRPM3, TRPA1, and TRPV1 in sensory neurons. By combining retrograde labeling and quantitative in situ hybridization using RNAscope, we obtained evidence suggesting an increase in mRNA encoding TRPM3 in dorsal root ganglion (DRG) neurons innervating the inflamed paw. Furthermore, we developed GCaMP3-based confocal imaging of DRG neurons in situ, which allowed us to measure inflammation-induced changes in the activity of TRPM3, TRPA1, and TRPV1, both in the neuronal cell bodies and in the intact nerve endings in the skin. These experiments indicate that tissue inflammation provokes a pronounced increase in the activity of nociceptors co-expressing TRPM3 with TRPV1 and TRPA1 channels. Notably, pharmacological inhibition of TRPM3 not only eliminated TRPM3-mediated responses but also reduced TRPV1- and TRPA1-mediated responses in DRG neurons innervating the inflamed paw. These findings elucidate a prominent role of TRPM3 in inflammatory hyperalgesia, and provide a rationale for the development of TRPM3 antagonists to treat inflammatory pain.

## Results

### Increased expression of TRPM3 mRNA in sensory neurons innervating inflamed tissue

To investigate the consequences of tissue inflammation on the expression and function of TRPM3, TRPA1, and TRPV1, we used a mouse model of complete Freund's adjuvant-induced (CFA-induced) peripheral inflammation. In this established model, unilateral hind paw injection of CFA produces a strong inflammatory response associated with pronounced hyperalgesia, and the contralateral hind paw can be used as an internal control.

First, we addressed whether tissue inflammation is associated with increased expression of mRNA encoding TRPM3, TRPA1, and TRPV1, in particular in sensory neurons that innervate the inflamed tissue. Sensory neurons innervating the hind paws of mice have their cell bodies in the dorsal root ganglia, primarily at the lumbar levels L3-L6. These ganglia also contain cell bodies of sensory neurons that innervate other parts of the body, including viscera, necessitating an approach to specifically label neurons that have endings in the hind paw. Therefore, we injected the retrograde label WGA-AF647 intraplantar in both hind paws seven days prior to tissue isolation, which resulted in effective labeling of cell bodies of sensory neurons that innervate the injected territory (*Figure 1A* and *Figure 1—figure supplement 1*). We induced an inflammatory response in the ipsilateral hindpaw by injecting CFA into the plantar surface 24 hr before tissue collection; at the same time, the contralateral hindpaw was injected with vehicle (*Figure 1—figure supplement 1*). We used single-molecule fluorescent RNA in situ hybridization (RNAscope) (*Wang et al., 2014*), to quantify the levels of mRNA encoding TRPM3, TRPA1 and TRPV1 in the ipsi- and contralateral DRGs (*Figure 1A*), and compared the levels of mRNA, both between retrogradely labeled (WGA-AF647$^+$) and unlabeled (WGA-AF647$^-$) neurons on the ipsilateral side, and between WGA-AF647$^+$ neurons on the ipsi- versus contralateral side. Importantly, this analysis revealed a significant increase in the TRPM3 mRNA levels in the ipsilateral, WGA-AF647$^+$ DRG neurons, both when compared to WGA-AF647$^+$ neurons

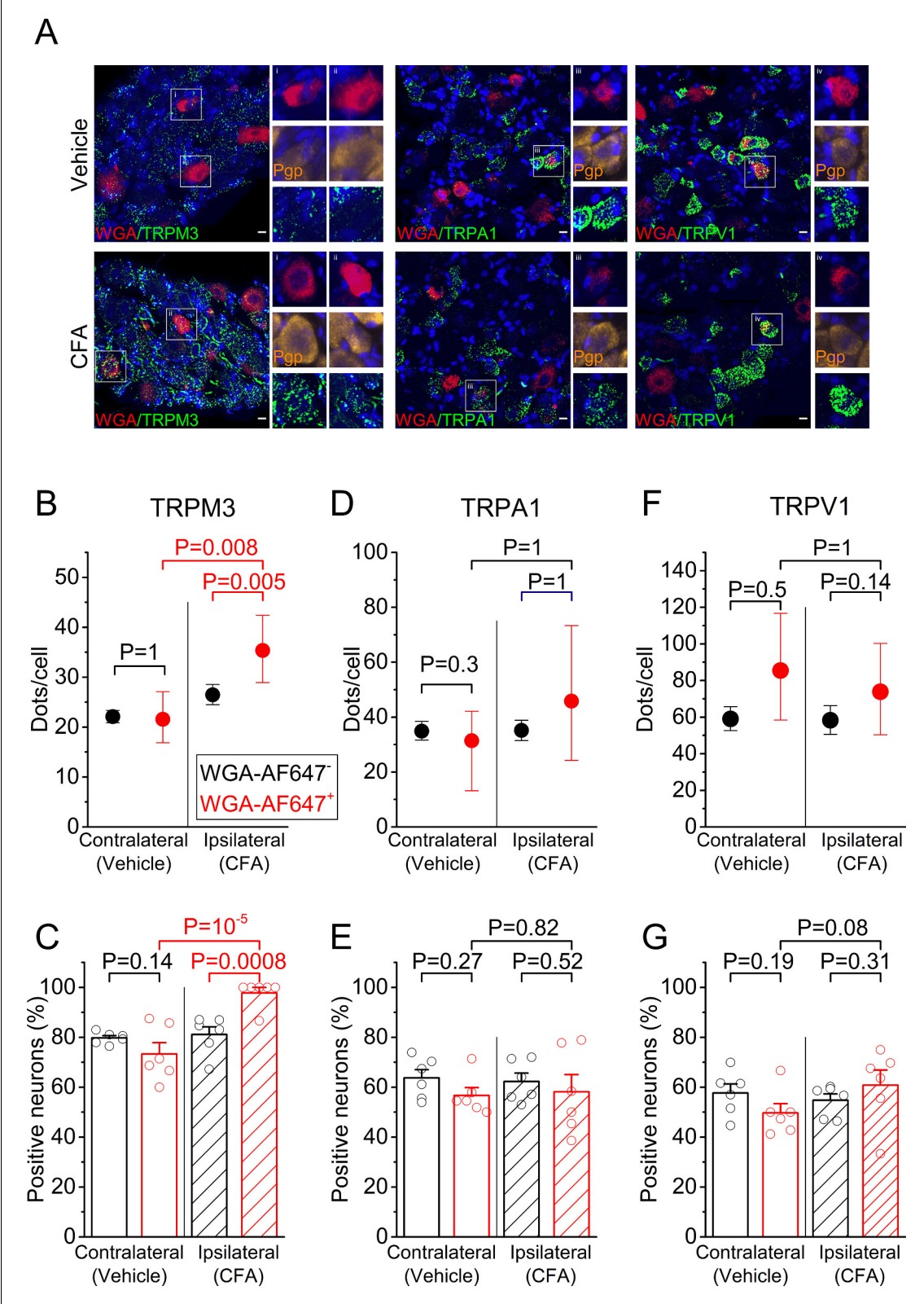

**Figure 1.** RNA expression of heat-activated TRP channels in sensory neurons innervating inflamed and control hind paws. (**A**) Representative fluorescent images of processed contralateral (vehicle-treated) and ipsilateral (CFA-treated) L5 DRG. Shown are RNAscope stainings with specific probes for TRPM3 (left), TRPA1 (middle) or TRPV1 (right) (green). Retrogradely labeled neurons were identified based on the WGA-AF647 staining (magenta), and the blue color represents the nuclear marker DAPI. Retrogradely labeled neurons in the boxed areas are shown at double magnification, along with RNAscope

*Figure 1 continued on next page*

*Figure 1 continued*

staining for the neuronal marker Pgp9.5 (yellow), which was used to delineate neuronal cell bodies. (B,D,F) Quantification of the number of RNAscope dots per DRG neuron for TRPM3, TRPA1 and TRPV1, comparing retrogradely labeled (red) and unlabeled (black) sensory neurons, from the contralateral and ipsilateral L5 DRG. Values are presented as mean along with the 95% confidence interval. The data of the individual cells are shown in *Figure 1—figure supplement 2*. Statistical comparisons between groups were made using Kruskal-Wallis ANOVA with Dunn's posthoc test. (C,E,G) Fraction of DRG neurons that showed a positive RNAscope signal ($\geq$5 dots) for the three tested channels. Values are presented as mean ± SEM along with data points from the individual mice. Statistical comparisons between groups were made using one-way ANOVA with Holm–Šidák post-hoc test. Data are from six mice. The total numbers of analyzed neurons were for TRPM3: 752 ipsilateral and 1299 contralateral; for TRPA1: 954 ipsilateral and 947 contralateral; for TRPV1: 1054 ipsilateral and 995 contralateral.

The online version of this article includes the following source data and figure supplement(s) for figure 1:

**Source data 1.** Raw values used for plots in *Figure 1*.
**Figure supplement 1.** Experimental setting: seven days before analysis, mice were injected bilaterally with the retrograde label WGA-AF647.
**Figure supplement 2.** Individual data points underlying the values shown in *Figure 1B,D,F* are displayed, along with box plots showing the median, first and third quartiles, whiskers showing the 5th and 95th percentiles, and open squares indicating the means.

on the contralateral side, and to WGA-AF647⁻ neurons on the ipsilateral side (*Figure 1B* and *Figure 1—figure supplement 2*). On average, TRPM3 mRNA levels in the ipsilateral, WGA-AF647⁺ DRG neurons were increased to 163% (95% confidence interval [CI], 122% to 222%; p=0.008) of the levels in the WGA-AF647⁺ neurons on the contralateral side, and to 133% (CI, 108–163%; p=0.005) of the levels in WGA-AF647⁻ neurons on the ipsilateral side. Notably, almost all (98 ± 2%) of the ipsilateral, WGA-AF647⁺ DRG neurons in the six tested animals showed a positive RNAscope signal for TRPM3, compared to between 70% and 80% TRPM3-positive neurons in the ipsilateral WGA-AF647⁻ and contralateral DRG neurons (*Figure 1C*).

In contrast, we did not observe significant inflammation-related changes in mRNA levels of TRPA1 or TRPV1 (*Figure 1D–G* and *Figure 1—figure supplement 2*). For TRPA1, mRNA levels in the ipsilateral WGA-AF647⁺ DRG neurons amounted to 120% (CI, 63% to 147%; p=0.56) of the level in the contralateral WGA-AF647⁺ DRG neurons and to 83% (CI, 40% to 139%; p=0.81) of the level in ipsilateral WGA-AF647⁻ neurons (*Figure 1D*). In the case of TRPV1, mRNA levels in the ipsilateral WGA-AF647⁺ neurons amounted to 89% (CI, 53% to 140%; p=0.52) of the level in the contralateral WGA-AF647⁺ DRG neurons and to 126% (CI, 87% to 166%; p=0.15) of the level in ipsilateral WGA-AF647⁻ neurons (*Figure 1F*). Likewise, the fraction of TRPA1- or TRPV1-positive DRG neurons did not differ significantly between WGA-AF647⁺ and WGA-AF647⁻ DRG neurons on the ipsilateral and contralateral sides (*Figure 1E,G*).

Taken together, these results suggest that inflammation is associated with an increased transcription of TRPM3, specifically in sensory neurons innervating the inflamed tissue, whereas no inflammation-related changes were found in the mRNA levels of TRPV1 and TRPA1.

## Increased TRP channel functionality in sensory neurons innervating inflamed tissue

Next, we investigated whether sensory neurons innervating inflamed tissue exhibit altered functionality of heat-sensitive TRP channels. In a first analysis, we focused on the functional expression of TRP channels in the neuronal cell bodies. Since procedures to isolate and culture DRG neurons significantly alter the expression levels of many ion channels (*Wangzhou et al., 2020*), we developed an assay where the DRG was imaged as a whole in situ, using spinning-disk confocal imaging (*Figure 2*). For these experiments, we made use of a mouse line that expresses the genetically encoded calcium sensor GCaMP3 in TRPV1-lineage neurons (TRPV1-GCaMP3 mice), which include all the neurons involved in thermosensation and inflammatory thermal hyperalgesia (*Mishra and Hoon, 2010*; *Mishra et al., 2011*; *Usoskin et al., 2015*). It is important to note that not all GCaMP3-positive DRG neurons from TRPV1-GCaMP3 mice functionally express TRPV1 at the developmental stage where we did our analysis. Indeed, in line with earlier studies (*Mishra et al., 2011*), we found that TRPV1-lineage neurons identified based on their GCaMP3 fluorescence include all DRG neurons that functionally express TRPV1 and TRPA1, but also a subset of DRG neurons that no longer expressed TRPV1 (*Figure 2—figure supplement 1*). Retrograde labeling using WGA-AF647 (see *Figure 1—figure supplement 1*) was used to identify the neurons that innervate the mouse paw. The efficiency of the retrograde labeling was similar in the ipsi- and contralateral side, with 25.8 ± 9.7% and 27.7 ±

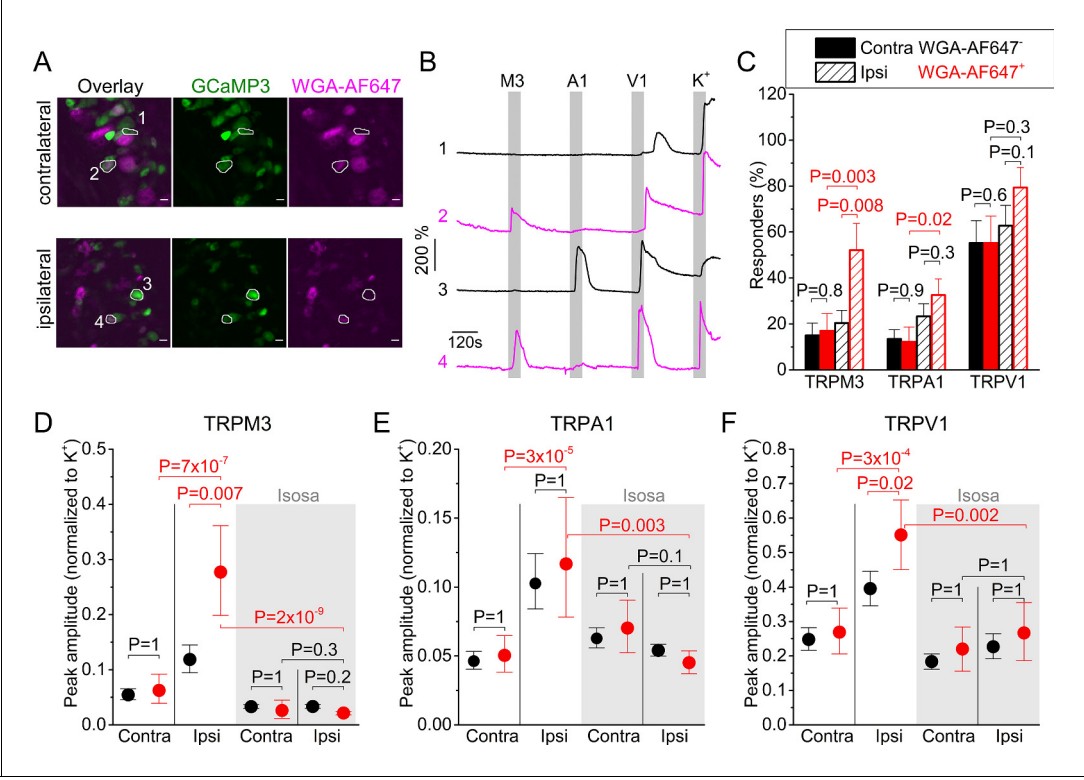

**Figure 2.** Inflammation-induced changes in TRP channel activity in DRG cell bodies. (**A**) Confocal images of the ipsi- and contralateral L5 DRG of a CFA-treated mouse showing GCaMP3 (green) and WGA-AF647 (magenta) fluorescence. (**B**) Representative examples, corresponding to cells indicated in panel (**A**) of changes in GCaMP3 fluorescence ($\Delta F/F_0$) in response to application of agonists of TRPM3 (PS + CIM0216; M3) TRPA1 (MO; A1) and TRPV1 (capsaicin; V1) and of a depolarizing high $K^+$ solution ($K^+$). (**C**) Percentage of neurons responding to the indicated agonists in WGA-AF647$^+$ (red) and WGA-AF647$^-$ (black) DRG neurons from the ipsi- and contralateral sides. Data are shown as mean ± SEM from 9 mice. Statistical comparisons between groups were made using one-way ANOVA with Holm–Šidák post-hoc test. Cells that did not respond to high $K^+$ stimulation were excluded from the analysis. (**D-F**) Peak amplitudes of responses to the TRPM3, TRPA1 and TRPA1 agonists, normalized to the response to the depolarizing high $K^+$ solution, comparing retrogradely labeled (WGA-AF647$^+$; red) and unlabeled (WGA-AF647$^-$; black) neurons on the ipsi- and contralateral side. Where indicated (grey background), DRGs were pre-incubated with isosakuranetin (20 µM). Values are presented as mean along with the 95% confidence interval. The data of the individual cells and the number of cells in the different groups are shown in *Figure 2—figure supplement 2*. A comparison of the non-normalized data is provided in *Figure 2—figure supplement 3*. Statistical comparisons between groups were made using Kruskal-Wallis ANOVA with Dunn's posthoc test. Data are from 9 mice in the absence of isosakuranetin and another set of 9 mice in the presence of isosakuranetin. The online version of this article includes the following source data and figure supplement(s) for figure 2:

**Source data 1.** Raw values used for plots in *Figure 2*.

**Figure supplement 1.** Combined GCaMP3- and Fura-2-based calcium imaging in isolated DRG neurons from TRPV1-GCaMP3 mice.

**Figure supplement 2.** Individual data points underlying the values shown in *Figure 2D–F* are displayed, along with box plots showing the median, first and third quartiles, whiskers showing the 5th and 95th percentiles, and open squares indicating the means.

**Figure supplement 3.** Non-normalized GCaMP3 responses.

**Figure supplement 4.** Percentage of the total imaged neurons responding to the indicated (combinations of) agonists in the contra- and ipsilateral DRG, both in control and following incubation with isosakuranetin (20 µM).

5.7% of the neurons being WGA-AF647$^+$, respectively. Changes in GCaMP3 fluorescence were monitored upon application of specific agonists for TRPM3 (the combination of pregnenolone sulfate; PS and CIM0216), TRPA1 (mustard oil; MO) and TRPV1 (capsaicin), and of a depolarizing high $K^+$ solution resulting in TRP channel-independent $Ca^{2+}$-influx via voltage-gated $Ca^{2+}$ channels (*Figure 2B*). The order of agonist application (TRPM3-TRPA1-TRPV1) and application timing was kept constant for all experiments. The order of agonist application was based on earlier experiments in isolated DRG neurons showing that capsaicin treatment leads to significant desensitization of sensory neurons to subsequent stimulation, whereas this is much less pronounced for the used agonists of TRPA1 and TRPM3 (*Vriens et al., 2011*; *Vandewauw et al., 2018*).

Both in the ipsi- and contralateral DRG, we observed calcium responses to the different agonist applications, which were indicative of neurons with different patterns of functional expression of one, two or all three heat-activated TRP channels (*Figure 2B*; *Figure 2—figure supplement 4*). We did not observe any significant differences in the fraction of neurons that responded to the different TRP channel agonists or in the response amplitudes between WGA-AF647$^+$ and WGA-AF647$^-$ neurons in the contralateral DRG, indicating that the retrograde label by itself did not affect TRP channel activity at the level of the cell bodies (*Figure 2C–F*; *Figure 2—figure supplements 2–4*). Likewise, responses to the depolarizing high K$^+$ solution did not differ significantly between groups, and these responses were to normalize the TRP channel-mediated responses (*Figure 2D–F*; *Figure 2—figure supplement 2*; non-normalized data are shown in *Figure 2—figure supplement 3*).

Importantly, responses to TRPM3 agonists were strongly increased in the ipsilateral WGA-AF647$^+$ neurons. The fraction of these neurons that exhibited a positive response increased significantly, to 52% compared to around 20% in the different control groups (*Figure 2C*). Likewise, the response amplitude in the ipsilateral WGA-AF647$^+$ neurons was increased to 233% ([CI], 157% to 330%; p=0.007) when compared to the ipsilateral WGA-AF647$^-$ neurons and to 448% ([CI], 265% to 755%; p=7 $\times$ 10$^{-7}$) when compared to the WGA-AF647$^+$ contralateral neurons (*Figure 2D*). These findings indicate a strong functional upregulation of TRPM3, specifically in neurons innervating the inflamed paw.

In the case of TRPA1, we found that the fraction of responding neurons in the ipsilateral WGA-AF647$^+$ neurons was doubled and the average response amplitude was increased to 229% ([CI], 146% to 355%; p=2 $\times$ 10$^{-5}$) when compared to the WGA-AF647$^+$ contralateral neurons (*Figure 2C, E*). However, there were no significant differences in the fraction of responding neurons or the response amplitudes between WGA-AF647$^+$ and WGA-AF647$^-$ neurons on the ipsilateral side, suggesting that increased responsiveness to TRPA1 agonists may not be limited to neurons innervating the hind paw but affects the entire ipsilateral DRG.

Finally, in the case of TRPV1, there was a trend toward more responders in the ipsilateral WGA-AF647$^+$ neurons, but this did not reach statistical significance (*Figure 2C*). However, the amplitude in the ipsilateral WGA-AF647$^+$ neurons was significantly increased, to 140% ([CI], 109% to 173%; p=0.02) when compared to the ipsilateral WGA-AF647$^-$ neurons and to 205% ([CI], 152% to 280%; p=3 $\times$ 10$^{-4}$) when compared to the WGA-AF647$^+$ contralateral neurons (*Figure 2F*). These findings indicate that the neurons that innervate the inflamed paw show enhanced responses to agonists of TRPM3, TRPA1, and TRPV1. Since these three channels show a partially overlapping expression profile (*Vandewauw et al., 2018*), we evaluated how tissue inflammation affects the functional co-expression of the three channels. Notably, in the ipsilateral WGA-AF647$^+$ neurons, we observed a pronounced increase in the fraction of neurons that show functional responses to agonists for both TRPM3 and TRPV1, as well as in the fraction that respond to all three agonists (*Figure 2—figure supplement 4*). Since these neurons showed a robust increase in mRNA expression of TRPM3, but not of TRPV1 and TRPA1 (*Figure 1*), we considered the possibility that increased TRPM3 expression and activity provoked by inflammation may contribute to enhanced excitability of sensory neurons, resulting in enhanced responses to TRPA1 and TRPV1 agonists. To investigate this possibility, we tested the responses of DRG neurons to TRP channel agonists in the presence of the TRPM3 antagonist isosakuranetin at 20 µM (*Straub et al., 2013*). This concentration is in accordance with the free plasma concentration of 17.5 ± 6.3 µM that is reached in mice following systemic (i.p.) application of 10 mg/kg isosakuranetin, a dose at which effective inhibition of TRPM3-mediated pain was reported (*Krügel et al., 2017*; *Straub et al., 2013*). As expected, isosakuranetin effectively eliminated the responses to TRPM3 agonists, illustrating the suitability of this compound to study TRPM3 functionality in DRG (*Figure 2D*; *Figure 2—figure supplements 2–4*). On the contralateral side, isosakuranetin had no significant effect on the responses to MO or capsaicin (*Figure 2E,F*), in line with the reported selectivity of the antagonist for TRPM3 (*Krügel et al., 2017*; *Straub et al., 2013*; *Jia et al., 2017*). Moreover, isosakuranetin did not affect the response to the depolarizing high K$^+$ solution, indicating that the compound does not Ca$^{2+}$ via voltage-gated Ca$^{2+}$ channels. Notably, isosakuranetin caused a significant inhibition of the response amplitudes to capsaicin and MO on the ipsilateral side. In fact, in the presence of isosakuranetin we no longer detected any significant differences in MO or capsaicin responses between ipsi- and contralateral, WGA-AF647$^+$, and WGA-AF647$^-$ neurons (*Figure 2E,F*; *Figure 2—figure supplements 2–4*).

Taken together, these results indicate that inflammatory heat hyperalgesia is associated with the increased functionality of all three heat-activated TRP channels at the level of the DRG cell bodies. In particular, we report for the first time a strong enhancement of TRPM3-mediated responses, as well as large increase in the fraction of neurons that co-express TRPM3 with TRPV1 and TRPA1. Notably, pharmacological inhibition of TRPM3 not only suppressed the responses to TRPM3 agonists but also reduced the responses to TRPA1 and TRPV1 agonists in neurons innervating inflamed tissue. These findings are in line with a model where increased molecular and functional expression of TRPM3 in the context of tissue inflammation enhances the excitability of sensory neurons, which may contribute to augmented responses of these neurons to TRPA1 and TRPV1 agonists.

## Optical measurements suggest increased TRP channel activity in cutaneous nerve endings in inflamed skin

Whereas these results indicate increased functionality of heat-activated TRP channels in the cell bodies of sensory neurons innervating the inflamed paw, they do not provide information regarding changes at the level of the sensory nerve endings. To address this issue, we developed an approach that allows direct measurement of TRP channel activity in intact cutaneous peripheral nerve endings in mouse hind paw skin (*Figure 3A*). In this assay, a skin flap of the dorsal surface of the hind paw and the innervating saphenous nerve (but lacking the DRG cell bodies) of TRPV1-GCaMP3 mice were excised and fixed in an organ bath, corium side up. Note that the plantar skin tissue is significantly thicker, and therefore less amenable for imaging nerve endings using this approach. Calcium signals in nerve endings expressing GCaMP3 were visualized from the epidermal side using a spinning-disk confocal microscope, while TRP channel agonists or a depolarizing high $K^+$ solution were locally applied from the dermal side (*Figure 3A*). We used a gravity-driven perfusion system with an outlet positioned on one border of the recording field, and constant suction on the opposite border, allowing rapid exchange of agonists. We adapted a published algorithm to automatically detect contiguous regions of interest (ROIs) exhibiting synchronous activity, which we interpret as individual branches of sensory nerve endings (*Zhou et al., 2018*). As illustrated in *Figure 3B,C* and *Figure 3—Video 1*, this approach revealed distinct calcium responses in localized regions, indicative of sensory endings that functionally express TRPM3, TRPA1, and TRPV1.

We used this novel approach to compare TRP channel activity in the skin of CFA- and vehicle-treated paws (*Figure 4A,B*). To minimize cross-(de)sensitization between stimuli, we used the same order of agonist applications as used in the DRG preparation (*Figure 2*), and allowed a wash-out period of 5 min between stimuli (*Figure 4A,B*). Since we measure responses from GCaMP3-positive nerve endings in the skin, not from clearly identifiable individual cells as in the DRG assay, a quantitative comparison of responses between ipsi- and contralateral paw posed several problems. First, the individual regions of interest that were automatically identified based on proximity and correlated activity represent only segments of individual sensory nerve endings, not the entire ending. Therefore, quantification of the percentage of nerve endings that respond to the specific TRP channel agonists was not feasible. Second, nerve endings in the skin did not always show robust responses to a depolarizing high $K^+$ solution, even when they showed robust responses to one or more TRP channel agonists (*Figure 4B*), making a normalization as was done for the neuronal cell bodies unreliable. The lack of consistent responses to a depolarizing high $K^+$ solution may reflect that not all nerve endings contain voltage-gated $Ca^{2+}$ channels or that inactivation of voltage-gated channels occurred due to the preceding TRP channel activation. Therefore, to quantify the functional expression of the three TRP channels, we determined the total area of automatically detected ROIs that showed a correlated response to each of the specific agonists, normalized this active area to the total imaged area (*Figure 4A*), and made a paired comparison between equivalent skin areas of the ipsi- and contralateral paws of the same animal. This analysis indicated increased reactivity to agonists of all three channels in the inflamed skin (*Figure 4C–E* and *Figure 4—figure supplement 1*). The responsive area was increased to 274% ([CI], 138% to 481%; p=0.04) for TRPM3 agonism, to 197% ([CI], 122% to 281%; p=0.02) for TRPA1 agonism and to 256% ([CI], 156% to 359%; p=0.02) for TRPV1 agonism, when compared to the contralateral side.

We also evaluated how tissue inflammation affects the functional co-expression of the three channels in the nerve terminals in the skin. Like in the DRG cell bodies, we observed a pronounced increase in the surface of nerve endings that showed responses to more than one agonist (*Figure 4F* and *Figure 4—figure supplement 1*). Notably, the increase in TRPM3-mediated responses was

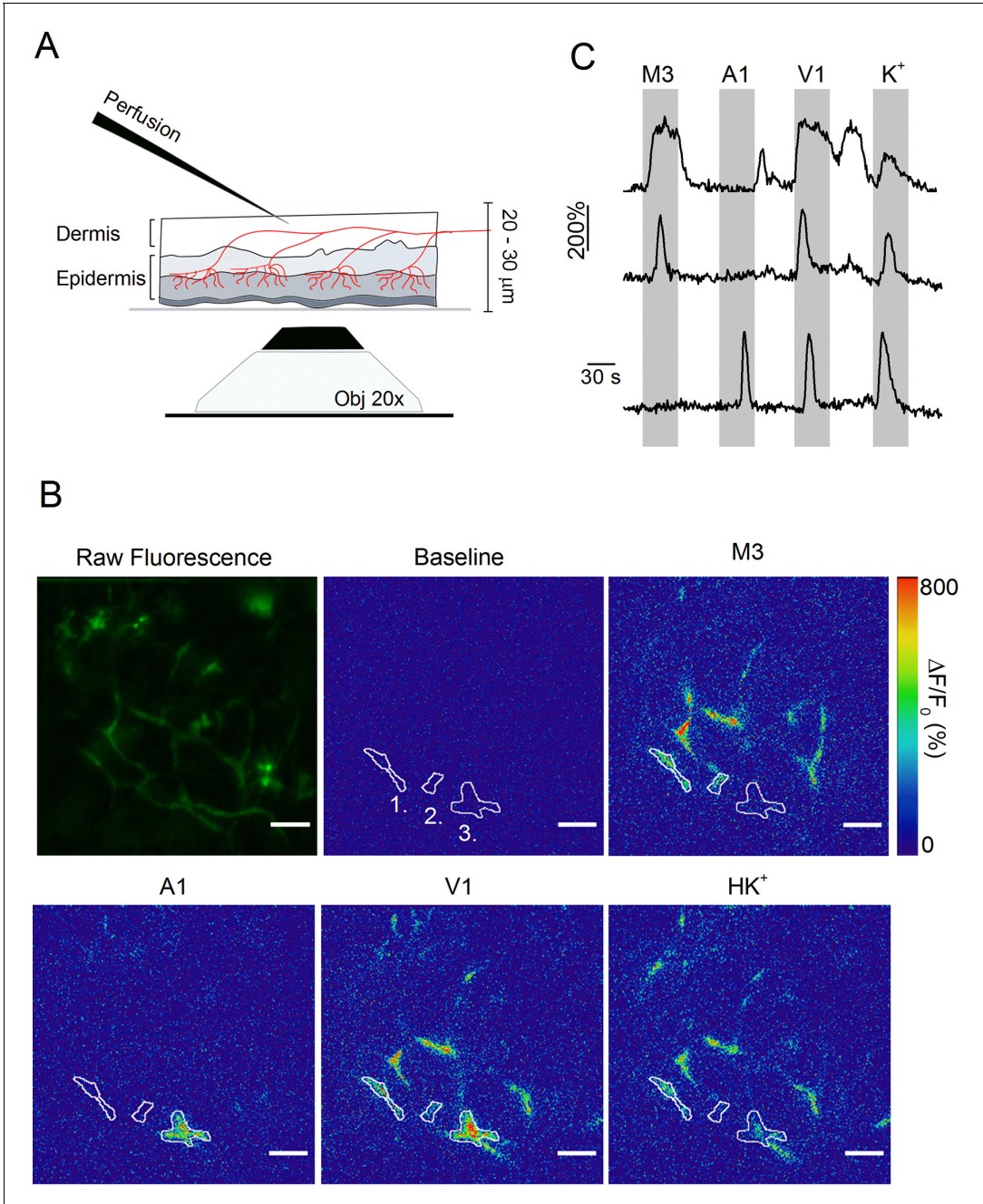

**Figure 3.** Optical measurement of TRP channel activity in peripheral sensory nerve endings. (A) Schematic illustration of the optical imaging setup. Sensory nerve fibers (red) innervating the dermal and epidermal skin layers are visualized using 488 nm laser light and an inverted spinning disk confocal microscope (20x objective). To avoid the barrier effect of the epidermis, solutions (at 37°C) were applied to the internal side of the sample from above. A total thickness of 20–30 μm was captured. (B) The first image depicts the summed raw fluorescence of the entire imaging experiment. The next five images represent normalized fluorescence (ΔF/F$_0$) at baseline (before the first stimulus), upon stimulation with TRPM3, TRPA1 and TRPV1 agonists, and with the depolarizing high K$^+$ solutions. Three automatically detected ROIs, corresponding to the traces in panel **C**, are indicated. Scale bar is 50 μm. See *Figure 3—Video 1*. (C) Time course of normalized GCaMP3 fluorescence (F/F$_0$) from three different ROIs (top: ROI 1; middle: ROI 2; bottom: ROI 3) depicted in panel **B**, with indication of the application periods of TRP channel agonists.

The online version of this article includes the following video for figure 3:

**Figure 3—video 1.** Video showing calcium-induced changes in GCaMP3 fluorescence in sensory nerve endings in mouse skin upon stimulation with TRP channel agonists.

https://elifesciences.org/articles/61103#fig3video1

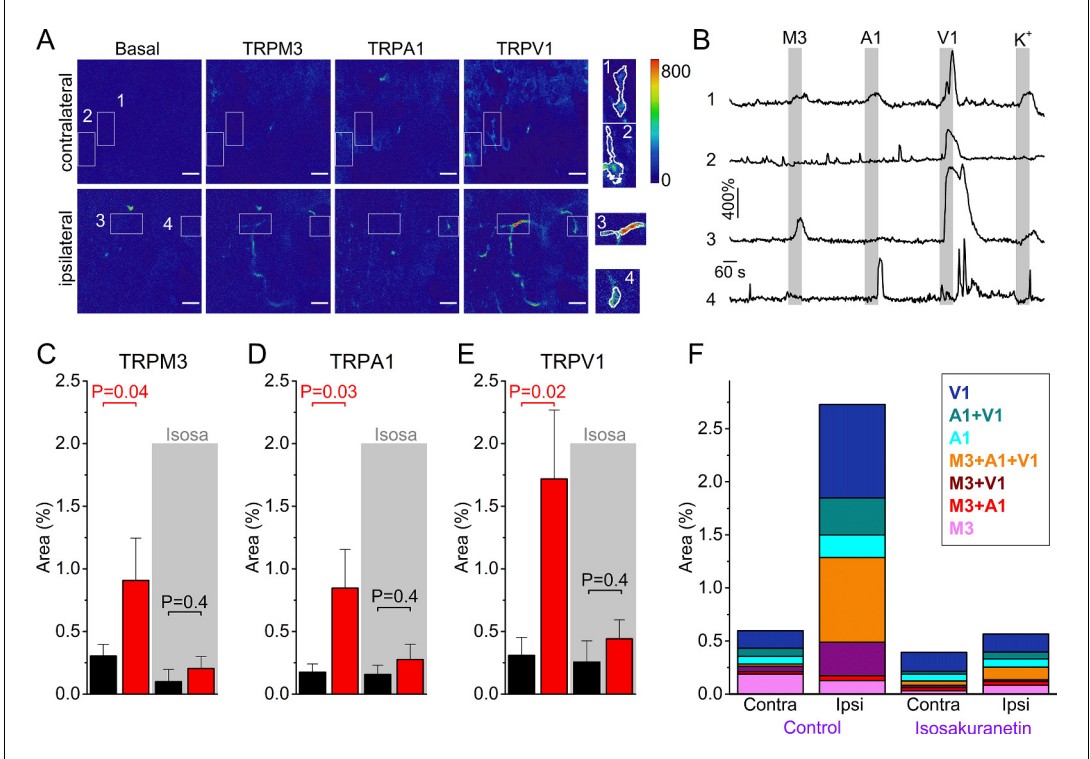

**Figure 4.** Increased TRPM3 activity in peripheral sensory nerve endings during inflammation. (**A**) Normalized fluorescence at baseline (before the first stimulus), and upon stimulation with TRPM3, TRPA1 and TRPV1 agonists of the ipsi- and contralateral skin of a CFA-treated mouse. Scale bar is 50 µm. Boxed areas, magnified on the right, illustrate automatically detected ROIs. (**B**) GCaMP3 fluorescence, expressed as ΔF/F$_0$, for the ROIs indicated in panel (**A**). (**C-E**) Responsive areas to the indicated agonists in the contralateral (black) and ipsilateral (red) skin. The paired Wilcoxon Signed Rank Test was used for a paired comparison of the responsive area in the ipsi- and contralateral paw skin of 11 mice, measured in the absence of isosakuranetin. The ipsi- and contralateral skin of another set of 6 mice was compared following pre-incubation with isosakuranetin (20 µM; grey background). Since the control and isosakuranetin-treated skin preparations originate from different mice, and considering the substantial inter-animal variability in skin thickness and innervation, a full statistical comparison between these data sets was not performed. (**F**) Percentage of the total imaged area responding to the indicated (combinations of) agonists in the contra- and ipsilateral paws, and following isosakuranetin pre-incubation. Further statistical comparison is provided in *Figure 4—figure supplement 1*.

The online version of this article includes the following source data and figure supplement(s) for figure 4:

**Source data 1.** Raw values used for plots in *Figure 4*.

**Figure supplement 1.** Inflammation-induced increases in sensory nerve endings that respond to multiple TRP agonists.

primarily observed in nerve endings that also responded to TRPV1 and/or TRPA1 agonists (*Figure 4F* and *Figure 4—figure supplement 1*). Overall, these qualitative results represent, to our knowledge, the first direct observation of functional upregulation of all three heat-activated TRP channels in intact nerve endings in inflamed skin.

In a separate set of experiments, we compared the TRP channel-mediated responses in the skin of the ipsi-and contralateral paws following pre-incubation with isosakuranetin. Under this condition, responses to TRPM3 agonists were strongly reduced to 12% ([CI], 3% to 45%; p=0.01; Mann-Whitney test) when compared to the ipsilateral paw skins measured in the absence of isosakuranetin. In these isosakuranetin-treated skin preparations, the surface of nerve endings responding to the TRPA1 and TRPV1 agonists were similar in the ipsilateral and contralateral skin (*Figure 4C–F*).

## Discussion

In recent work, we demonstrated that three TRP channels (TRPM3, TRPV1, and TRPA1) act as redundant sensors of acute heat: neuronal heat responses and heat-induced pain are preserved in mice in which two of these three TRP channels were genetically ablated, but combined elimination of all three channels fully abolishes the withdrawal reflex from a noxious heat stimulus (*Vandewauw et al.,*

*2018*). Notably, earlier work had also revealed an absolute requirement of both TRPV1 and TRPM3 for the development of inflammatory heat hypersensitivity, since genetic ablation or pharmacological inhibition of either channel individually fully abrogates heat hyperalgesia in the rodent CFA model (*Caterina et al., 2000*; *Davis et al., 2000*; *Gavva et al., 2005*; *Honore et al., 2005*; *Alkhatib et al., 2019*; *Krügel et al., 2017*; *Vriens et al., 2011*). These combined findings suggested that inflammation modulates the functional interplay between these heat-activated TRP channels leading to pathological heat hypersensitivity, but the underlying processes and mechanisms remained unclear. Here, by directly comparing the levels of mRNA expression and functional activity of TRPM3, TRPV1 and TRPA1 between sensory neurons innervating the healthy and inflamed hind paw in mice, we arrive at the following novel conclusions: (1) acute inflammation is associated with an increased TRPM3 expression at the mRNA level, specifically in DRG neurons innervating the inflamed paw; no significant changes were detected in the mRNA levels for TRPA1 or TRPV1; (2) inflammation is associated with increased functionality of TRPM3, TRPA1 and TRPV1 in DRG neurons, as evidenced by increased $Ca^{2+}$ responses to specific agonists; this increased functionality is detected both in the neuronal cell bodies and in the nerve endings in the inflamed skin; (3) increased TRP channel activity in inflammatory conditions is associated with an increase in the fraction of cell bodies and nerve endings that functionally co-express TRPM3 with TRPV1 and TRPA1, and (4) pharmacological inhibition of TRPM3 not only eliminates $Ca^{2+}$ responses to TRPM3 agonists, but also reduces responses to TRPV1 and TRPA1 agonists in neurons innervating the inflamed paw. Taken together, these data demonstrate, for the first time, an increased molecular and functional expression of TRPM3 in nociceptors innervating inflamed tissue, and indicate that tissue inflammation is associated with a functional upregulation of all three molecular heat sensors implicated in the normal pain response to noxious heat.

Whereas this is, to our knowledge, the first study pointing at possible inflammation-induced alterations in TRPM3 expression at the mRNA level, earlier studies had already investigated alterations in TRPA1 and TRPV1 mRNA using the rodent CFA model. In line with our current results, several earlier studies report that transcript levels for TRPV1 were found unchanged after CFA treatment when assessed using quantitative PCR or RNA protection assays (*Endres-Becker et al., 2007*; *Ji et al., 2002*). However, some earlier studies found increased levels of TRPA1 mRNA in lumbar DRG following CFA treatment of the hind paw (*Obata et al., 2005*; *Zhou et al., 2013*). These results seem at odds with our current findings, which did not reveal a statistically significant increase in TRPA1 mRNA in neurons innervating the inflamed paw. It should be noted, however, that in these earlier studies, bulk mRNA from entire ganglia was analyzed, which inevitably includes not only neurons that innervate injured tissue, but also neurons from the same DRG that innervate healthy tissue, as well as non-neuronal cells present in DRG such as satellite glia (*Shin et al., 2020*). In contrast, our approach using quantitative in situ hybridization and retrograde labeling allowed us to specifically measure mRNA levels in the cell bodies of the sensory neurons that innervate the inflamed or control hind paw. On the other hand, given the significant variability in TRPA1 mRNA levels between individual DRG neurons observed in our RNA-scope experiments, and the relatively limited number of neurons that were identified as WGA-AF647$^+$, we consider the possibility that an overall increase in TRPA1 mRNA in the order of 30% or less would not be revealed in our statistical analysis.

We used two novel approaches to assess inflammation-induced alterations in the functionality of the three TRP channels, one at the level of the cell bodies and the other at the level of the sensory nerve endings of DRG neurons. First, we developed an acute ex vivo preparation of the relevant part of the spinal column, where the cell bodies of sensory neurons were imaged within an intact DRG, but lacking the peripheral input. This preparation avoids the potential loss of specific neuronal populations as well as rapid alterations in their functional properties that are inherent to the isolation and culturing of DRG neurons (*Wangzhou et al., 2020*). The use of the TRPV1-cre line to drive GCaMP3 expression allowed the specific recording from neurons involved in thermosensation and nociception (*Mishra and Hoon, 2010*; *Mishra et al., 2011*), whereas the use of WGA-AF647-labeling ensured the specificity for afferent neurons that innervated the hind paw tissue. With the use of specific agonists, we were able to directly and quantitatively compare the TRPM3-, TRPA1- and TRPV1-induced activity between cell bodies of neurons that innervated the inflamed versus the control paw. Interestingly, we found increased responses to agonists for all three channels and a particular increase in neurons that functionally co-expressed TRPM3 with TRPV1 and TRPA1. Given the stringent criterium we used to classify a cell as WGA-AF647$^+$ (see methods), it is not unlikely that a

subset of DRG neurons that innervate the inflamed paw did not take up sufficient dye to be considered as retrogradely labeled. This may explain why we also observed a mild increase in TRPM3 mRNA (*Figure 1B*) and TRP-mediated responses (*Figure 2D–F*) in the ipsilateral, WGA-AF647⁻ DRG neurons. Increased activity of TRPV1 and TRPA1 in DRG neurons following CFA-induced inflammation is fully in line with earlier work (*Zhou et al., 2013*; *Breese et al., 2005*; *Nicholas et al., 1999*), but this study is, to our knowledge, the first to reveal the specific increase in the subpopulation of DRG co-expressing the three heat-sensing TRP channels. Secondly, we developed an ex vivo assay, where we used GCaMP3-based calcium imaging to monitor TRP channel-mediated responses in intact sensory nerve endings in the paw skin. The skin preparation that we used is identical to the saphenous skin–nerve preparation that has been widely studied using extracellular electrodes to measure propagated action potentials from the receptive fields of single sensory nerve endings in the skin, including responses evoked by TRP channel agonists such as capsaicin (*Reeh, 1986*; *Zimmermann et al., 2009*). This novel imaging approach allowed us to demonstrate for the first time that TRPM3, TRPA1, and TRPV1 show a pattern of partial spatial functional overlap in peripheral nerve endings in the skin. Moreover, similar to the findings in the neuronal cell bodies, our results suggest an increase in the surface of nerve endings that functionally express TRPM3 in the skin of the inflamed paw, particularly in endings that also respond to TRPV1 and TRPA1 agonists.

The potent TRPM3 antagonist isosakuranetin effectively eliminated TRPM3-mediated calcium responses, both in the DRG cell bodies and nerve terminals. Intriguingly, isosakuranetin also tempered the responses to capsaicin and MO in DRG neurons innervating the inflamed paw, restoring both the response amplitudes and the fraction of responding cells to the same level as in neurons innervating uninjured tissue. We can exclude the possibility that the reduced responses are due to a direct inhibitory effect of isosakuranetin on TRPV1 or TRPA1 activity since the responses to capsaicin and MO were unaffected by the TRPM3 antagonists in neurons innervating healthy tissue. Likewise, earlier studies have shown that isosakuranetin does not inhibit heterologously expressed TRPV1 or TRPA1 (*Straub et al., 2013*). Instead, these data raise the possibility that increased molecular and functional expression of TRPM3 in neurons innervating inflamed tissue increases the excitability of nociceptors co-expressing TRPA1 and TRPV1, contributing to the augmented responses to agonist stimulation. This interpretation also provides a straightforward mechanism for the observation that heat hyperalgesia does not develop in TRPM3-deficient mice and is fully alleviated by TRPM3 antagonists (*Alkhatib et al., 2019*; *Vriens et al., 2011*). Since heat hyperalgesia is also strongly attenuated by pharmacological inhibition or genetic ablation of TRPV1 (*Caterina et al., 2000*; *Davis et al., 2000*; *Gavva et al., 2005*; *Honore et al., 2005*), we hypothesize that those DRG neurons that gain functional co-expression of TRPM3 and TRPV1 under inflammatory conditions play a central role in the development of heat hypersensitivity.

In conclusion, the present findings provide the first evidence that TRPM3 expression and activity are increased in sensory neurons that innervate acutely inflamed tissue. In particular, we found a marked increase in sensory neurons that functionally co-express TRPM3 with TRPV1 and TRPA1, two other TRP channels implicated in heat sensing and inflammatory pain, and this was observed both in the peripheral nerve endings and in the DRG cell bodies. Strikingly, pharmacological inhibition of TRPM3 also reduced TRPV1- and TRPA1-mediated responses in nociceptors innervating the inflamed paw but not in the contralateral control neurons, suggesting that enhanced TRPM3 activity may contribute to neuronal hyperexcitability under inflammatory conditions. Therefore, these results provide a straightforward rationale for the development of TRPM3 antagonists to prevent or alleviate inflammatory pain.

## Materials and methods

**Key resources table**

| Reagent type (species) or resource | Designation | Source or reference | Identifiers | Additional information |
|---|---|---|---|---|
| Genetic reagent (*M. musculus*) | C57BL/6JRj | Janvier Labs | | https://www.janvier-labs.com/en/fiche_produit/c57bl-6jrj_mouse/ |

*Continued on next page*

*Continued*

| Reagent type (species) or resource | Designation | Source or reference | Identifiers | Additional information |
|---|---|---|---|---|
| Genetic reagent (*M. musculus*) | TRPV1-GCaMP3 | This paper | | Obtained by crossing Gt(ROSA)26S or$^{tm38(CAG-GCaMP3)Hze}$/J mice (Stock#: 029043) with Trpv1$^{tm1(cre)Bbm}$ mice (Stock#: 017769). Both strains were acquired from Jackson Laboratory. Crossings were made in house. |
| Peptide, recombinant protein | WGA-AF647 (wheat germ agglutinin- Alexa Fluor 647) | Thermo Fisher Scientific | Cat#: W32466 | 0.8% in PBS; 10 µl per injection |
| Commercial assay or kit | RNAscope 2.0 Fluorescent Multiplex Reagent Kit | Advanced Cell Diagnostics | Cat#: 320850; RRID:SCR_012481 | |
| Commercial assay or kit | TRPV1 Probe | Advanced Cell Diagnostics | Cat#: 313331 | |
| Commercial assay or kit | TRPM3 Probe | Advanced Cell Diagnostics | Cat#: 459911 | |
| Commercial assay or kit | TRPA1 Probe | Advanced Cell Diagnostics | Cat#: 400211 | |
| Commercial assay or kit | PgP9.5 Probe | Advanced Cell Diagnostics | Cat#: 561861-C2 | |
| Chemical compound, drug | isosakuranetin | Extrasynthese | Cat#: 1374 | 20 µM |
| Chemical compound, drug | CIM0216 | Sigma-Aldrich | Cat#: 534359 | 1 µM |
| Chemical compound, drug | Pregnenolone sulfate | Sigma-Aldrich | Cat#: P162 | 100 µM |
| Chemical compound, drug | Mustard oil | Sigma-Aldrich | Cat#: W203408 | 100 µM |
| Chemical compound, drug | capsaicin | Sigma-Aldrich | Cat#: M2028 | 1 µM |
| Chemical compound, drug | DAPI | Thermo Fisher Scientific | Cat#: P36931 | |
| Software, algorithm | Turboreg algorithm | https://imagej.net/TurboReg | RRID:SCR_003070 | |
| Software, algorithm | CNMF-E algorithm | https://github.com/zhoupc/CNMF_E | RRID:SCR_001622 | |
| Software, algorithm | NIS software | Nikon Instruments | RRID:SCR_014329 | |
| Software, algorithm | OriginPro 2019b | Originlabs | RRID:SCR_014212 | |
| Software, algorithm | Igor Pro 8 | Wavemetrics | | |

*Continued on next page*

*Continued*

| Reagent type (species) or resource | Designation | Source or reference | Identifiers | Additional information |
| --- | --- | --- | --- | --- |
| Other | CFA (complete freund's adjuvant) | Sigma-Aldrich | Cat#: F5581 | (1 mg/ml) 10 µl per injection |
| Other | Glass-bottom microwell dish | MatTek | Cat#: P35G-1.5–14 C | |
| Other | Glass-bottom chamber | Fluorodish, WPI | Cat#: FD35-100 | |
| Other | DAPI | Thermo Fisher Scientific | Cat#: P36931 | |

## Animals

C57BL/6J wild type mice (Janvier Labs, Le Genest-Saint-Isle, France) and TRPV1-GCaMP3 mice on a C57BL/6J background were used. TRPV1-GCaMP3 mice were generated by crossing Rosa26-floxed-GCaMP3 mice (*Zariwala et al., 2012*) with TRPV1-cre mice (*Mishra et al., 2011*). Mice were housed in a conventional facility at 21°C on a 12 hr light-dark cycle with unrestricted access to food and water. Both male and female mice between 8 and 12 weeks of age were used. Experiments were performed in concordance with EU and national legislation and approved by the KU Leuven ethical committee for Laboratory Animals under project number P075/2018 and P122/2018.

## Reagents

Reagents were purchased from Sigma-Aldrich (Chemical Co., St. Louis, Missouri) unless otherwise indicated.

## Retrograde labeling

To specifically label afferents from the hind paw skin, 10 µl Alexa Fluor 647-conjugated wheat germ agglutinin (WGA-AF647, Thermo Fisher Scientific, Invitrogen, Eugene, Oregon, USA; 0.8% in sterile PBS) was injected intraplantar into both hind paws. The injections were performed 7 days prior to imaging. Initial experiments showed no edematous or hyperalgesic response to the retrograde label.

## Paw inflammation

Local inflammation was induced by injection of 10 µl complete Freund's adjuvant (CFA, 1 mg/ml) into the plantar surface of the ipsilateral hindpaw of the studied mouse. The contralateral hind paw was injected with 10 µl vehicle (saline, Baxter, Lessen, Belgium). All ipsilateral mouse hind paws showed substantial edema 24 hr after the injection, which was not observed in the control paw.

## Calcium imaging

Animals were euthanized using $CO_2$ inhalation, and skin and DRG tissue were collected immediately.

### Skin nerve preparation

Sensory nerve ending recordings were obtained from isolated dorsal hind paw skin preparations. The fur was removed with tape and the skin was gently dissected from the underlying tissue.

### In situ DRG preparation

Bilateral L3-L6 DRGs were isolated. In brief, the spinal column was isolated, cleaned, and split sagittally. The spinal cord, meninges covering the DRG, and the distal axon bundles were removed. Finally, a small segment of the spinal column containing the DRG of interest was extracted.

The isolated skin tissue and spinal column segments were maintained 1 hr on ice and 30 min at room temperature in synthetical interstitial fluid (SIF) solution, containing (mM): 125 NaCl, 26.2 $NaHCO_3$, 1.67 $NaH_2PO_4$, 3.48 KCl, 0.69 $MgSO_4$, 9.64 D-gluconic acid, 5.55 D-glucose, 7.6 Sucrose

and 2 $CaCl_2$. The pH was buffered to 7.4 using carbogen gas (95% $O_2$ and 5% $CO_2$). In some experiments, the latter 30 min incubation solution as well as the proceeding perfusion SIF solution contained TRPM3 blocker isosakuranetin (20 µM; Extrasynthese, Genay Cedex, France). During image acquisition, skin tissue was fixed with the corium side up in a glass-bottom microwell dish (MatTek, 35 mm petri dish, Ashland, USA). The spinal column sections were placed in the SIF-containing microwell dishes with the DRG facing towards the objective. Tissues were continuously superfused with 37°C SIF solution.

Calcium imaging of the skin-nerve and in situ DRG preparations was performed on an inverted spinning disk confocal microscope (Nikon Ti; Yokogawa CSU-X1 Spinning Disk Unit, Andor, Belfast, Northern Ireland), equipped with a 20x air objective (NA 0.8), a 488 nm laser light and a EMCCD camera (iXon3 DU-897-BV, Andor). For image acquisition and instrument control, Andor iQ software was used. A z-stack of 11 frames (total thickness of 20–30 µm) was captured consecutively during the entire measurement at a speed of 0.25 Hz. Agonists were diluted in SIF and applied using a heated perfusion system (Multi Channel Systems, Reutlingen, Germany). Before image acquisition, the DRG samples were excited at 640 nm to detect WGA-647$^+$ retrogradely labeled cell bodies. The used stimuli to activate the specific TRP channels were for TRPM3: PS (100 µM) + CIM0216 (1 µM); for TRPA1: MO (100 µM); and for TRPV1 capsaicin (1 µM). Compounds were diluted in SIF supplemented with 0.1% DMSO. A depolarizing high K$^+$ solution in which all NaCl was replaced by KCl was applied at the end of each experiment to identify excitable cells. All solutions were applied at 37°C and administered to the immediate proximity of the recording field.

### Isolated DRG neurons
In a limited set of experiments, shown in *Figure 2—figure supplement 1*, we performed combined Fura-2 and GCaMP3 fluorescence measurements on isolated DRG neurons. Ganglia were excised, washed in neurobasal A medium (Invitrogen) supplemented with 10% fetal calf serum (basal medium), and then incubated for 45 min at 37°C in a mix of 1 mg/ml collagenase and 2.5 mg/ml dispase (Gibco). Digested ganglia were gently washed twice with basal medium and mechanically dissociated by passage through syringes fitted with increasing needle gauges. Neurons were seeded on poly-L-ornithine/laminin coated glass-bottom chambers (Fluorodish, WPI) and cultured overnight at 37°C in 5% $CO_2$ in B27 (Invitrogen) supplemented neurobasal A medium, containing 2 ng/ml GDNF (Invitrogen) and 10 ng/ml NT4 (Peprotech). Isolated DRG neurons were loaded with 2 µM Fura-2-acetoxymethyl ester (Alexis Biochemicals) for 30 min at 37°C. Fluorescence was measured during alternating illumination at 340 and 380 nm (yielding the Fura-2 fluorescence ratio) and at 480 nm (to monitor GCaMP3 fluorescence) using a Cell$^M$ (Olympus) fluorescence microscopy system.

## Image processing and analysis
Spinning disk confocal microscope recordings were first z-averaged in Fiji (imageJ, 1.52i) and corrected for translational drift using a reference image that averaged the first 10 frames prior to tissue stimulation (Turboreg plugin, imageJ).

### Skin nerve
After drift correction, we used custom-made routines in Igor (Wavemetrics) to subtract fluorescence background. The background was obtained by fitting a fourth-order polynomial surface to non-responsive areas, which were defined as those pixels where the coefficient of variation in time was below a threshold, which was automatically obtained following the Otsu algorithm. After background correction, we used the 'constrained non-negative matrix factorization for micro-endoscopic data' framework (CNMF-E, Matlab R2017b) to identify ROIs as contiguous areas with temporally correlated activity (*Zhou et al., 2018*). The background-subtracted fluorescence was normalized to the basal fluorescence of the reference image (first 10 frames prior to stimulation), yielding $F/F_0$ values.

### In situ DRG
These recordings were analyzed using a general analysis protocol (GA2) in NIS-elements (NIS 5.20.00, Nikon Instruments Europe B.V.). Retrogradely labeled (WGA-AF647$^+$) and unlabeled (WGA-AF647$^-$) neurons were initially identified visually; for analysis, only cells with a fluorescence signal that exceeded five times the standard deviation of the background fluorescence in this channel were

retained in as WGA-AF647$^+$. Raw fluorescent traces were converted to F/F$_0$, where F is the fluorescence at a certain time point of interest, and F$_0$ is the baseline fluorescence before stimulation. Cells that did not respond to high K$^+$ where excluded from all further analysis.

To determine responders in the skin-nerve and DRG recordings, two criteria had to be fulfilled: first, the peak F/F0 during stimulation had to exceed five times the standard deviation of the F/F0 value before stimulation. Second, the peak of the first derivative of the fluorescent signal (dF/dt) had to exceed the standard deviation of the dF/dt signal before stimulation.

### In situ hybridization

In situ hybridization was performed on 10-µm-thin cryosections of DRG neurons innervating the contralateral and ipsilateral hind paw. DRG tissue was isolated as described previously and immersed in 10% neutral buffered formalin immediately after isolation. Prior to isolation, retrograde labeling and inflammation were induced as described previously. RNA transcripts were detected using the RNAscope 2.0 assay according to the manufacturer's instructions (Advanced Cell Diagnostics, Hayward, CA, United States). Probes for mTrpv1 (cat number: 313331), mTrpm3 (cat number: 459911), mTrpa1 (cat number: 400211) and Pgp9.5 (cat number: 561861-C2) were purchased from Advanced Cell Diagnostics. The staining was performed using the RNAscope Fluorescent Multiplex Reagent Kit (cat number: 320850). Cells were stained with DAPI and mounted on the slide with Gold Antifade Mountant. A total of 5802 neurons were analyzed from 12 lumbar (L5) DRG neurons (6 contralateral and six ipsilateral), isolated from six mice.

Slides were imaged using a Nikon NiE - Märzhäuser Slide Express two equipped with a Hamamatsu Orca Flash 4.0 in combination with a Plan Apo 40x (NA 0.95), and custom made JOBS-GA2 protocol for sample detection. For analysis, a GA3 script in NIS-Elements 5.20.00 was used. Cells were segmented manually based on the Pgp 9.5 signal. Cells were considered as WGA-AF647$^+$ if the mean fluorescence exceeded five times the standard deviation of the background fluorescence in this channel. Due this stringent criterium, it is likely that some retrogradely labeled neurons with modest WGA-AF647 fluorescence are included in the WGA-AF647$^-$ group. Individual dots in the green channel were detected using a rolling ball filter (1 µm) and spot detection (0.8 µm). To correct for dot clusters arising from the overlap of individual RNAscope-dots, we determined the average intensity of single dots in each slide, and calculated the theoretical number of individual dots in a cluster as the ratio of the cluster intensity and the average intensity of single dots, as outlined in the manufacturers instructions (https://acdbio.com/ebook/introduction/materialsmethod). Cells containing five or more RNAscope dots were considered as positive for the respective TRP channels.

### Statistical analysis

Data analysis was performed using Origin software (OriginPro 2019b). Shapiro–Wilk test was used to test the normality of the data, determining whether parametric or non-parametric tests were applied. The specific parametric and non-parametric tests that were used are specified in the text and legends. Summary data for parametric datasets are shown as mean + SEM. We used bootstrapping with 10,000 bootstrap resamples to calculate 95% confidence intervals for non-parametric datasets. No statistical methods were used to predetermine the number of animals used in this study; since no a priory data were available for the novel imaging approaches used in this work, a power calculation was not feasible. However, our sample sizes are similar to those generally employed in other studies in the field, and potential limitations due to insufficient power are discussed in the text. Imaging of the DRG and skin preparations and subsequent data analysis were performed by researchers that were blinded for the treatment (vehicle *versus* CFA).

## Acknowledgements

We acknowledge the Cell and Tissue imaging cluster (CIC; KU Leuven), where confocal microscopy was performed, and the Light Microscopy and Imaging Network (LiMoNe; CBD-VIB), where in situ hybridization slides were imaged. Calcium imaging at CIC was performed on an Andor Revolution Spinning Disk System supported by Hercules AKUL/09/50 to PVB. This research was further supported by grants from the VIB, KU Leuven Research Council (C1-TRPLe to TV), the Research Foundation-Flanders (FWO G0B7620N to TV), the Belgian Foundation Against Cancer (to JV and TV) and the Queen Elisabeth Medical Foundation for Neurosciences (to TV).

## Additional information

### Funding

| Funder | Grant reference number | Author |
|---|---|---|
| Hercules Foundation | AKUL/09/50 | Pieter Vanden Berghe |
| KU Leuven | C1-TRPLe | Thomas Voets |
| Fonds Wetenschappelijk Onderzoek | G0B7620N | Thomas Voets |
| Belgian Foundation Against Cancer | | Joris Vriens<br>Thomas Voets |
| Queen Elisabeth Medical Foundation for Neurosciences | | Thomas Voets |
| Vlaams Instituut voor Biotechnologie | | Sebastian Munck<br>Thomas Voets |
| Fonds Wetenschappelijk Onderzoek | G0B5316N | Thomas Voets |

The funders had no role in study design, data collection and interpretation, or the decision to submit the work for publication.

### Author contributions

Marie Mulier, Conceptualization, Data curation, Formal analysis, Funding acquisition, Investigation, Writing - original draft, Writing - review and editing; Nele Van Ranst, Formal analysis, Investigation; Nikky Corthout, Software, Formal analysis, Investigation, Writing - review and editing; Sebastian Munck, Supervision, Methodology, Writing - review and editing; Pieter Vanden Berghe, Methodology, Writing - review and editing; Joris Vriens, Supervision, Writing - review and editing; Thomas Voets, Conceptualization, Data curation, Software, Formal analysis, Supervision, Funding acquisition, Investigation, Methodology, Writing - original draft; Lauri Moilanen, Conceptualization, Formal analysis, Supervision, Investigation, Methodology, Writing - original draft, Writing - review and editing

### Author ORCIDs

Marie Mulier  https://orcid.org/0000-0001-8429-7136
Nele Van Ranst  https://orcid.org/0000-0003-2242-4050
Sebastian Munck  https://orcid.org/0000-0002-5182-5358
Pieter Vanden Berghe  https://orcid.org/0000-0002-0009-2094
Joris Vriens  http://orcid.org/0000-0002-2502-0409
Thomas Voets  https://orcid.org/0000-0001-5526-5821

### Ethics

Animal experimentation: Experiments were performed in concordance with EU and national legislation and approved by the KU Leuven ethical committee for Laboratory Animals under project number P075/2018 and P122/2018.

### Decision letter and Author response

Decision letter https://doi.org/10.7554/eLife.61103.sa1
Author response https://doi.org/10.7554/eLife.61103.sa2

## Additional files

### Supplementary files
• Transparent reporting form

### Data availability

All data points generated during this study are included in the manuscript and figures.

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
