## [Decision Letter]

**Acceptance summary:**

Mulier and colleagues investigate the molecular mechanisms that underlie the development of increased sensitivity to heat pain after inflammation. Their interest focuses on the ion channel TRPM3 that this group has previously shown is thermosensitive and contributes to heat sensation. Here, the authors investigate whether the expression of TRPM3 in sensory neurons is altered during peripheral inflammation and whether its function contributes to hyperalgesia. They authors provide evidence that TRPM3 message is upregulated in the sensory neurons innervating inflamed tissues using a combination of retrograde tracing and single molecule florescent in situ hybridization (FISH). They also use calcium imaging in ex vivo preps (whole ganglia and skin) to examine the function of TRPM3 and two other TRP channels important for temperature and pain signaling (TRPV1 and TRPA1) using pharmacology. Overall, their approach is technically innovative and the concept that TRPM3 is required for increased TRPV1 and TRPA1 activity is intriguing.

**Decision letter after peer review:**

Thank you for choosing to send your work, "Upregulation of TRPM3 drives hyperexcitability in nociceptors innervating inflamed tissue", for consideration at *eLife*. Your article has been reviewed by two peer reviewers, and the evaluation has been overseen by Kenton Swartz as the Senior Editor and Reviewing Editor. Although the work is of interest, we regret to inform you that the findings at this stage are too preliminary for further consideration at *eLife*.

As you will see, both reviewers thought your study was exciting and both thought your skin preparation had real potential. However, both reviewers thought that more needed to be done in terms of controls, quantification and providing more details, in particular for the new preparation. Although there was considerable enthusiasm for your work, the policy at *eLife* is to only invite revisions if the additional work can be done within 2 month. We encourage you to consider the reviewers comments carefully, and if you choose to address all of their comments, we would be willing to reconsider your work as a new submission.

Reviewer #1:

In their current manuscript, Mulier et al. investigate the molecular mechanisms that underlie the development of increased sensitivity to heat pain after inflammation. Their interest focuses on the ion channel Trpm3 that this group has previously shown is thermosensitive and contributes to heat sensation. Here, they ask whether the expression of Trpm3 in sensory neurons is altered during peripheral inflammation and whether its function contributes to hyperalgesia. They authors provide evidence that Trpm3 message is upregulated in the sensory neurons innervating inflamed tissues using a combination of retrograde tracing and single molecule florescent in situ hybridization (FISH). They also use calcium imaging in ex vivo preps (whole ganglia and skin) to examine the function of TRPM3 and two other Trp channels important for temperature and pain signaling (TRPV1 and TRPA1) using pharmacology.

Overall, their approach is technically innovative and the ideas interesting. While it is clear Trp channels contribute to heat and pain sensitization, it is still unclear their precise roles and interactions. Thus, the concept that TRPM3 is required for increased TRPV1 and TRPA1 activity is really intriguing. However, the manuscript title and claims that the upregulation of TRPM3 is necessary and sufficient to cause nociceptor hyperexcitability are too bold. The data show a correlation between elevated TRPM3 expression and functional TRP channel responses in inflamed tissue, but a causal relationship is not well-established. Moreover, the robustness of the initial premise that TRPM3 is upregulated in inflamed tissue is somewhat debatable. How TRPM3 might be affecting TRPV1 or TRPA1 function is really unclear. Finally, I have some issues regarding how the RNAscope and skin calcium imaging experiments were quantified.

Given the effect sizes are pretty small, these issues need addressing before publication in *eLife*. I also recommend reducing the strength of the claims and that including additional analyses would be helpful.

Figure 1:

– In Figure 1B, is the Pgp9.5 labeling via antibody or RNAscope? This is really unclear from the figure legend and methods. No product number was listed in Methods.

– To accompany Figure 1B, it would help to show an in situ image representing differences between control and CFA-treated TRPM3 neurons. Only control example images are shown

– In Figure 1B, why does the TRPV1 in situ staining look much sparser than expected based on published gene expression analyses? Typically, Trpv1 ISH labels many small diameter neurons quite strongly, where are these cells?

– In Figure 1C, the method of FISH quantification is non-standard. Typically for single molecule FISH, the number of puncta per cell are quantified (as detailed in the RNAscope paper cited in the Results section), not the overall fluorescence intensity of an ROI. These data would be more accurate if reanalyzed using a standard "puncta counting" approach.

– Furthermore, difference between animals (not cells) should be shown.

– Overall, the data in Figure 1 are not very convincing without a re-analysis (as described in the above point). A secondary method to confirm the FISH results would greatly strengthen the data, such as using FACS isolation and qRT-PCR analysis of TRP genes in WGA-positive neurons.

– Why do the ipsilateral WGA-negative neurons in the Figure 1C FISH and the Figure 2D Ca^2+^ imaging appear to have elevated TRP gene expression and responses compared to the contralateral WGA-negative neurons? While I can see that the WGA labeling is not perfect, it seems a bit odd that the entire ipsilateral DRG is inflamed, not just the paw-innervating neurons.

Figure 2:

– The data for WGA-negative cells should also be shown

– In general, normalization to high potassium is still not sufficient to make very detailed comparisons between cells without using a ratiometric calcium probe. Normally, this isn't an issue and many in vivo studies rely on GCaMP, but here the effect sizes are small enough to warrant caution.

– In Figure 2D-F, data from the Iso treatment experiments should additionally be shown for WGA-negative neurons (for a total of 4 conditions per side). This is needed to demonstrate that the TRPM3 antagonist doesn't affect TRPA1 and TRPV1 responses at baseline in the neurons that don't innervate the inflamed tissue. This seems to be a particularly critical control.

– In the bar plots, the Y-axes scales are each different and some have break lines. Things should be shown on the same scale to make comparisons across graphs easier

– I recognize performing new experiments under the current conditions would be tough, but it is strange they don't look at heat responses, since this is what the title and Abstract of the paper are about

Figures 3 and 4:

– Figure 3 really is an example of the skin imaging technique, which is I think very cool. However, I am not convinced about the quantification in Figure 4. They admit that they cannot normalize to the maximum response. I don't see how they can account for the natural variation between different skin preparations (size, thickness density of innervation, health, etc). Overall, I'm not convince this shows what the authors claim- the effects are very small and the spread of the data quite large.

At the end of the section, the authors write: "Taken together, these results indicate that inflammatory heat hyperalgesia is associated with increased functional expression of all three heat-activated TRP channels at the level of the DRG cell bodies." But they only show increased expression of TRPM3 in Figure 1.

Reviewer #2:

The authors have recently demonstrated that heat-induced pain in naïve mice depends on a trio of heat-activated TRP channels, TRPM3, TRPA1 and TRPV1. However, with tissue inflammation, genetic ablation or pharmacological inhibition of only TRPV1 or only TRPM3 inhibition can fully suppress inflammatory heat hyperalgesia. Here the authors seek to dissect the relative contributions of TRPM3, TRPA1 and TRPV1 to nociceptor sensitization during tissue inflammation.

In this manuscript, the authors use RNAscope to demonstrate that there is an increase in TRPM3 expression 1 day following CFA injection in the hind paw, specifically in the neurons that innervate the hind paw; there is no difference in TRPA1 or TRPV1 expression. Using whole DRG calcium imaging and a novel ex vivo skin calcium imaging approach, they reveal that neurons are more sensitive to TRPV1 and TRPA1 agonists following CFA hind paw injections and that this sensitivity is dependent on TRPM3 activity; V1 and A1 hypersensitivity can be blocked with the M3 antagonist isosakuranetin. Based on these experiments, TRPM3 antagonists may be viable therapeutic candidates for heat hypersensitivity. The paper is generally well written and accessible to a broad audience.

• The authors claim that M3-dependent TRPA1 hypersensitivity occurs following CFA but neither the amplitude of TRPA1 responses (seems as if just 2 outliers are driving this difference; Figure 2) nor the percentage of neurons responding to TRPM3+TRPA1 are very supportive of this claim. Furthermore, the increase in A1 sensitivity does not appear to be specific to paw-innervating (i.e. WGA-labeled) neurons, because the same increase (based on a few outliers) occurs in non-paw innervating unlabeled neurons. While the skin imaging data are more convincing, the authors should dampen their claim that TRPM3 expression increases TRPA1 sensitivity following injury.

• Since TRPV1-GCaMP3 mice are being used for both DRG and skin calcium imaging, how is it possible that not all cells respond to TRPV1 agonists? In other words, how are M3 only, A1 only, or M3+A1 responders detectable using this genetic line since only TRPV1 positive neurons would express GCaMP3? It seems that this would limit the authors' ability to detect changes in A1 sensitivity following injury. If this population of TRPV1-lineage neurons includes all of the TRPM3 and TRPA1-expressing neurons in the adult, then this should be made even more clear in the manuscript.

• The authors should discuss whether there is potential cross desensitization/sensitization between agonist in both the DRG and the peripheral skin calcium imaging experiments, since cross-interactions between agonists could limit the strength of their conclusions.

• More information/citations should be provided as to the dose/concentration chosen for the TRPM3 antagonist isosakuranetin, particularly since a single concentration was used of this single antagonist.

• The authors are commended for ex vivo imaging of sensory afferent terminals in the skin. Depending on the scientific question, this approach could have many benefits over imaging exclusively at the level of the DRG. Since this is the first description of this method, the authors should further clarify the methods in detail and address the following questions:

How large is the bath containing the skin? How does buffer exit the bath? It appears that agonists are being presented to the tissue for 30-45 s. with an equivalent washout period between compounds (Figure 3C). Is this sufficient time for the agonists to both access sensory terminals deep in the skin and then be washed away from the tissue? More details are needed on whether there is potential desensitization from one agonist to the next if insufficient time is allowed for agonist washout. More details on this method, the regions of interests chosen and how the data are quantified in a blinded manner should be provided.

• The authors should comment on the use of a saphenous (presumably dorsal hairy skin) preparation use in the context of the current injury model (i.e. 10 µL injection of CFA to the plantar surface of the paw)?

I don't think that the use of the contralateral paw is the best control since there could be segmental effects, and separate animal control would be better. That said, this would require the entire data set to be re-generated and I'm not going to recommend this. But this is something that I think that the authors should note going forward and avoid using the contralateral paw as the control.

[Editors’ note: further revisions were suggested prior to acceptance, as described below.]

Thank you for submitting your article "Upregulation of TRPM3 in nociceptors innervating inflamed tissue" for consideration by *eLife*. Your article has been reviewed by two peer reviewers, and the evaluation has been overseen by Kenton Swartz as the Senior Editor and Reviewing Editor. The following individual involved in review of your submission has agreed to reveal their identity: Cheryl L Stucky (Reviewer #1).

The reviewers have discussed the reviews with one another and the Reviewing Editor has drafted this decision to help you prepare a revised submission.

The editors and reviewers found the revised manuscript to be greatly improved, and we appreciate your sincere efforts to address the concerns that were raised. Overall, the conclusions better align with the results obtained, although we have some remaining concerns with conclusions concerning the RNAscope and skin prep results.

RNA scope:

The example images in Figure 1 are improved, however the new staining for Trpm3 looks to either be quite non-specific or to have high background? Either might affect the conclusions. In addition, the reviewers remain unconvinced that TRPM3 expression increases in response to inflammation. Some changes might seem big percent-wise but cell to cell variation and the biological meaning of such differences should also be considered: for example, what's the difference between an average of 20 and 30 puncta when the range of variation between cells is far greater? While these results may be suggestive, our consensus is the RNAscope results are not conclusive. We were much more convince by the increased functional responses to TRPM3 agonists. We request that you tone down conclusions concerning the RNAscope results throughout the manuscript. For example, in the Results section the following sentence:

“Taken together, these results indicate that inflammation is associated with a significantly increased transcription of TRPM3, specifically in sensory neurons innervating the inflamed tissue, whereas no inflammation-related changes were found in the mRNA levels of TRPV1 and TRPA1.”

could be easily improved by replacing 'indicate' with 'suggest' and "a significantly" with 'an'.

Skin assay:

The limitations of the assay are clearly spelled out in the Results section, but as analyzed the results are inherently qualitative. As such we suggest also toning down the conclusion here. For example, in the Results the following sentence:

“Overall, these results represent, to our knowledge, the first direct observation of functional upregulation of all three heat-activated TRP channels in intact nerve endings in inflamed skin.”

could be improved by inserting 'qualitative' before 'results' to remind the reader of the nature of the measurements.

---

## [Author Response]

[Editors’ note: the authors resubmitted a revised version of the paper for consideration. What follows is the authors’ response to the first round of review.]

Reviewer #1:In their current manuscript, Mulier et al. investigate the molecular mechanisms that underlie the development of increased sensitivity to heat pain after inflammation. Their interest focuses on the ion channel Trpm3 that this group has previously shown is thermosensitive and contributes to heat sensation. Here, they ask whether the expression of Trpm3 in sensory neurons is altered during peripheral inflammation and whether its function contributes to hyperalgesia. They authors provide evidence that Trpm3 message is upregulated in the sensory neurons innervating inflamed tissues using a combination of retrograde tracing and single molecule florescent in situ hybridization (FISH). They also use calcium imaging in ex vivo preps (whole ganglia and skin) to examine the function of TRPM3 and two other Trp channels important for temperature and pain signaling (TRPV1 and TRPA1) using pharmacology.Overall, their approach is technically innovative and the ideas interesting. While it is clear Trp channels contribute to heat and pain sensitization, it is still unclear their precise roles and interactions. Thus, the concept that TRPM3 is required for increased TRPV1 and TRPA1 activity is really intriguing.However, the manuscript title and claims that the upregulation of TRPM3 is necessary and sufficient to cause nociceptor hyperexcitability are too bold. The data show a correlation between elevated TRPM3 expression and functional TRP channel responses in inflamed tissue, but a causal relationship is not well-established.

We feel that this is based on a misunderstanding. At no point in the manuscript did we intend to claim that the upregulation of TRPM3 is necessary *and sufficient* to cause nociceptor hyperexcitability, and we agree that such a statement would be too bold at this point. Maybe it is because none of us is a native English speaker, but we did not interpret our statements that "TRPM3 is an important/key driver of nociceptor hyperexcitability excitability" to be synonymous to "TRPM3 is necessary and sufficient to cause nociceptor hyperexcitability".

What we meant is that TRPM3 is an important contributor. We felt that such a statement was justified based on the novel findings that (1) TRPM3 mRNA expression is specifically upregulated, (2) TRPM3 function is robustly increased, particularly in neurons co-expressing TRPV1 and TRPA1, and (3) inhibition of TRPM3 function restores TRPV1- and TRPA1-mediated responses to normal levels, along with earlier findings that TRPM3 KO or pharmacological inhibition largely abolished hypersensitivity. We have reworded the title to something more unambiguous: “Upregulation of TRPM3 in nociceptors innervating inflamed tissue.”, and also revised all other instances where it could be implied that TRPM3 upregulation is sufficient to cause neuronal hyperexcitability. We do not think that these changes change the conclusions or importance of our manuscript.

Moreover, the robustness of the initial premise that TRPM3 is upregulated in inflamed tissue is somewhat debatable.

We feel that our initial way of presenting the data, showing all individual data points, may have led the reviewers to underestimate the actual magnitude of the TRPM3 upregulation. As outlined in detail below, the effect size of the effect is very robust and highly significant.

How TRPM3 might be affecting TRPV1 or TRPA1 function is really unclear.

In our Discussion, we propose a scenario whereby increased molecular and functional expression of TRPM3 in neurons innervating inflamed tissue increases the excitability of nociceptors co-expressing TRPA1 and TRPV1, contributing to the augmented responses to agonist stimulation. As such, less TRPV1 or TRPA1 activity would be sufficient to cause neuronal action potentials, leading to enhanced neuronal responses to agonists. We also discuss how this could explain that both TRPM3 knockout and TRPV1 knockout mice do not develop inflammatory hyperalgesia. At this point, we do not have any evidence for a direct molecular interaction between TRPM3 and the two other channels, but also do not exclude such a possibility.

Finally, I have some issues regarding how the RNAscope and skin calcium imaging experiments were quantified.

As outlined below, these issues have been resolved with additional analyses and further clarification of our methods.

Given the effect sizes are pretty small, these issues need addressing before publication in eLife.

See below. We politely disagree that the effect sizes are "*pretty small*." For instance, concerning TRPM3, we observe a >60% increase in mRNA (with P values as low as 10^-5^) and >300% increase in functional responses (P<10^-6^). We also reanalyzed the data sets according to the different suggestions of this reviewer (per animal instead of per cell; with or without normalization to high potassium responses; see the revised Figures 1 and 2, and their figure supplements), and the results remain very similar and highly significant.

From this and later comments, including also one comment from reviewer #2, we now realize that the way of representing our data (showing all individual data points and diamond plots) was maybe not the best choice to visualize the magnitude of the effects (although it is a recommended way of showing non-parametric data). By including additional summary plots and analyses, we are convinced that the reviewer will appreciate the size and robustness of our main conclusions.

I also recommend reducing the strength of the claims and that including additional analyses would be helpful.

We have carefully revised all the claims that we make in the manuscript and modified them whenever they could be perceived as too strong and not fully supported by the data.

Figure 1:– In Figure 1B, is the Pgp9.5 labeling via antibody or RNAscope? This is really unclear from the figure legend and Materials and methods. No product number was listed in Materials and methods.

We apologize for this omission. Pgp9.5 labeling was also done via RNAscope. This is now indicated in the Materials and methods.

– To accompany Figure 1B, it would help to show an in situ image representing differences between control and CFA-treated TRPM3 neurons. Only control example images are shown

We now provide better images, showing both the vehicle- and CFA-treated conditions.

– In Figure 1B, why does the TRPV1 in situ staining look much sparser than expected based on published gene expression analyses? Typically, Trpv1 ISH labels many small diameter neurons quite strongly, where are these cells?

Thank you for pointing this out. There is indeed a large number of neurons that are intensely labeled by the TRPV1 RNAscope label, much more prominent than for TRPM3 and TRPA1. However, we agree that the original image of the in situ staining did not show this clearly, due to inappropriate contrast. This is now much more evident in the provided images in the new Figure 1A. In the original manuscript, the strongly labeled cells could be appreciated from the summary data, showing cells with very high dot intensity values compared to TRPM3 and TRPA1. In the new manuscript, Figure 1—figure supplement 2 shows cells with up to 1000 TRPV1 RNAscope dots, consistent with the literature.

– In Figure 1C, the method of FISH quantification is non-standard. Typically for single molecule FISH, the number of puncta per cell are quantified (as detailed in the RNAscope paper cited in the Results section), not the overall fluorescence intensity of an ROI. These data would be more accurate if reanalyzed using a standard "puncta counting" approach.

This is a misunderstanding based on a too brief explanation of the analysis method in the original manuscript, for which we apologize. We did not analyze the overall fluorescence of a ROI, but only the summed intensity of all spots after subtraction of non-specific background. This quantification is equivalent to the number of spots but also corrects for the fact that brighter/larger spots likely arise from superimposed/overlapping RNAscope dots. However, we understand that a puncta counting approach is the standard, and now report the number of dots as the prime parameter. This does not alter any of our conclusions.

– Furthermore, difference between animals (not cells) should be shown.

We agree that it is important to also look at differences between animals. However, given that the number of dots per cell is distributed in a highly non-Gaussian manner, including many negative cells, important information would get lost if we would only show a mean or median value per animal. Therefore, we now include two analyses in the manuscript. First, we provide a statistical comparison of the number of dots/cell for all analyzed cells (ipsilateral and contralateral, retrogradely labeled or not) from the different mice (Figure 1B, D, F and Figure 1—figure supplement 2). In addition, we also calculated the percentage of positive neurons (with a cutoff of 5 RNAscope dots) for each condition per mouse (Figure 1C, E, G). Both analyses point at a robust and highly significant increase in TRPM3 mRNA expression in the neurons innervating the inflamed paw.

– Overall, the data in Figure 1 are not very convincing without a re-analysis (as described in the above point). A secondary method to confirm the FISH results would greatly strengthen the data, such as using FACS isolation and qRT-PCR analysis of TRP genes in WGA-positive neurons.

As outlined above, we believe that the additional analyses and statistics provide convincing evidence for a robust increase in TRPM3 mRNA expression. In the text, we now additionally provide a clear indication of the size of the effects, including confidence intervals and P values. As we now write: "On average, TRPM3 mRNA levels in the ipsilateral, WGA-AF647+ DRG neurons were increased to 163% (95% confidence interval [CI], 122% to 222%; P=0.008) of the levels in the WGA-AF647+ neurons on the contralateral side, and to 133% (CI, 108%-163%; P=0.005) of the levels in WGA-AF647- neurons on the ipsilateral side."

While FACS isolation and qPCR could, in principle, be a valuable approach, our experiences indicate that there is an unequal survival and representation of the different subtypes of DRG after the dispersion of the neurons and FACS sorting. Moreover, the procedure to isolate and disperse the neurons may by itself, influence mRNA expression levels. Therefore, we believe that our quantitative in situ hybridization approach is at this point, superior. Furthermore, the increased functional responses to TRPM3 agonists (Figures 2-4) provide further independent support for the increased expression of TRPM3.

– Why do the ipsilateral WGA-negative neurons in the Figure 1C FISH and the Figure 2D Ca^2+^ imaging appear to have elevated TRP gene expression and responses compared to the contralateral WGA-negative neurons? While I can see that the WGA labeling is not perfect, it seems a bit odd that the entire ipsilateral DRG is inflamed, not just the paw-innervating neurons.

We used a relatively conservative fluorescence threshold for a neuron to be considered WGA-positive. Therefore, there is likely a subset of neurons that innervate the injured paw but contain only a low level of WGA-AF647, and are therefore nevertheless included in the WGA-negative group. This is now stated in the Materials and methods section. Moreover, the CFA-induced inflammation causes massive inflammation in the entire hind paw, which may include regions where little or no WGA-AF647 was present. Therefore, it can be expected that part of the ipsilateral, WGA-negative neurons also innervate the inflamed paw, which may explain why we see a mild increase in mRNA and functional responses in these cells. For these reasons, we consider the WGA-positive neurons on the contralateral side as a cleaner control. Finally, it cannot be excluded that inflammation has some effects on neighboring DRG neurons within the same ganglion that are not innervating the inflamed paw. We now briefly discuss these issues in the revised manuscript.

Figure 2:– The data for WGA-negative cells should also be shown

These data are now included. In fact, Figure 2 has been completely revised, including two figure supplements.

– In general, normalization to high potassium is still not sufficient to make very detailed comparisons between cells without using a ratiometric calcium probe. Normally, this isn't an issue and many in vivo studies rely on GCaMP, but here the effect sizes are small enough to warrant caution.

From this comment, we realize that the way of representing our data (showing all individual data points and diamond plots) was not fully adequate to visualize the magnitude of the effects, and we have included additional plots to make the difference more evident. In the revised manuscript, we now show summary data in the main Figure 2: the percentage of responders (based on data from individual mice) in Figure 2C, and the mean amplitudes (with indication of 95% CI) for the different groups in Figure 2D-F. Data from individual cells are still provided, but now in Figure 2—figure supplement 2. In addition, in the text we provide actual values (and 95% confidence intervals) for the relative increase in response amplitude for the different agonists in the ipsilateral, WGA-positive neurons compared to controls. We are confident that, with this way of representing the data, it becomes obvious that the effect sizes are not small by any means. For instance, the response amplitude to TRPM3 agonist in the ipsilateral WGA-positive neurons was increased to 448% ([CI], 265% to 755%; P=7×10^-7^) when compared to the WGA-positive contralateral neurons (Figure 2D). Also for TRPA1 and TRPV1, we measured at least a doubling of the average response amplitude when compared to the WGA-positive contralateral neurons.

As to the normalization to high potassium, we have reanalyzed and compared the responses both with and without normalization. First, we found that there was no difference in absolute response amplitudes to the high potassium stimulus between the groups, as is now shown in Figure 2—figure supplement 3A. Next, we found that by and large the reported differences between the groups remain significant when normalization was not applied, as is now shown in Figure 2—figure supplement 3B, D. We note, however, that normalization to high potassium reduced the overall variation, and thus that most P-values are slightly higher when no normalization is applied. We are confident that, by showing both the normalized and non-normalized data, the readers can readily appreciate the robustness of the described effects.

– In Figure 2D-F, data from the Iso treatment experiments should additionally be shown for WGA-negative neurons (for a total of 4 conditions per side). This is needed to demonstrate that the TRPM3 antagonist doesn't affect TRPA1 and TRPV1 responses at baseline in the neurons that don't innervate the inflamed tissue. This seems to be a particularly critical control.

These data are now included in Figures 2D-F and in the figure supplements.

– In the bar plots, the Y-axes scales are each different and some have break lines. Things should be shown on the same scale to make comparisons across graphs easier

Thank you for the suggestions. We have made new summary plots in Figure 2 that make a direct comparison much easier.

– I recognize performing new experiments under the current conditions would be tough, but it is strange they don't look at heat responses, since this is what the title and Abstract of the paper are about

We note that heat is not mentioned in the title. The primary aim of the manuscript was to look at alterations in the expression and function of known heat-activated TRP channels. However, the use of pharmacological tools to probe for their activity is certainly more specific, because a heat stimulus will evoke a mixed response with potential contributions of all three channels and as well as additional specific and non-specific signals. We have tried extensively to use heat as a stimulus in the ex vivo preparations, but this was not at all straightforward. First, using heat as a stimulus in the ex vivo skin preparation during imaging generally caused local movements, which made it difficult to reliably track responses in the small nerve endings. Moreover, genetically encoded calcium indicators such as GCaMP3 show substantial temperature dependence, with both reduced calcium binding affinity (due to faster off rates) and lower fluorescence upon heating (see e.g. https://www.nature.com/articles/srep15978), thereby significantly reducing the amplitude of heat-evoked signals. While we are currently considering alternative approaches that would allow measurement of heat-evoked responses in these preparations (e.g. using different activity indicators), we consider it beyond the scope of the present manuscript, which was aimed at showing changes in expression and function of TRPM3, TRPA1 and TRPV1.

Figures 3 and 4:– Figure 3 really is an example of the skin imaging technique, which is I think very cool. However, I am not convinced about the quantification in Figure 4. They admit that they cannot normalize to the maximum response. I don't see how they can account for the natural variation between different skin preparations (size, thickness density of innervation, health, etc). Overall, I'm not convince this shows what the authors claim- the effects are very small and the spread of the data quite large.

In this assay, we directly compared the skin from the two hind paws of individual mice (we used a paired test, as indicated in the legend), thereby accounting for natural differences in skin thickness and innervation density between animals. Skins were prepared by researchers with yearlong experience in using this preparation for electrophysiological recordings, and maintained in a standardized manner before recording. The size of the imaged area was identical for all experiments. The experiments and their analysis was performed without knowledge of whether the tissue was from the ipsi- or contralateral side. We included additional information on the technique in the Materials and methods section of the revised manuscript.

Although there is significant spread, we have to politely but strongly disagree that the effects are very small or not convincing. The average increase is >200% for all three channels! We now provide actual values (along with a 95% confidence interval) for the relative increase in the ipsilateral versus contralateral side in the manuscript, allowing a more direct assessment of the magnitude of the effect. The results are statistically significant, and precise P-values are indicated in the figure. We now also show results from individual experiments in Figure 4—figure supplement 1. With these robust and statistically significant outcomes, we believe it is fully justified to make the claims that we made, namely "*increased reactivity to agonists of all three channels in the inflamed skin*".

At the end of the section, the authors write: "Taken together, these results indicate that inflammatory heat hyperalgesia is associated with increased functional expression of all three heat-activated TRP channels at the level of the DRG cell bodies." But they only show increased expression of TRPM3 in Figure 1.

"*Functional expression*" of ion channels/transporters is a term that we and others frequently use to denote the effective function of these proteins (e.g. current, transport) at the membrane. Also here, we meant to point at the increased functional responses, which are evident for all three channels (Figures 2 and 4). We have changed this to "*increased functionality*" to avoid any misunderstanding.

Reviewer #2:The authors have recently demonstrated that heat-induced pain in naïve mice depends on a trio of heat-activated TRP channels, TRPM3, TRPA1 and TRPV1. However, with tissue inflammation, genetic ablation or pharmacological inhibition of only TRPV1 or only TRPM3 inhibition can fully suppress inflammatory heat hyperalgesia. Here the authors seek to dissect the relative contributions of TRPM3, TRPA1 and TRPV1 to nociceptor sensitization during tissue inflammation.In this manuscript, the authors use RNAscope to demonstrate that there is an increase in TRPM3 expression 1 day following CFA injection in the hind paw, specifically in the neurons that innervate the hind paw; there is no difference in TRPA1 or TRPV1 expression. Using whole DRG calcium imaging and a novel ex vivo skin calcium imaging approach, they reveal that neurons are more sensitive to TRPV1 and TRPA1 agonists following CFA hind paw injections and that this sensitivity is dependent on TRPM3 activity; V1 and A1 hypersensitivity can be blocked with the M3 antagonist isosakuranetin. Based on these experiments, TRPM3 antagonists may be viable therapeutic candidates for heat hypersensitivity. The paper is generally well written and accessible to a broad audience.• The authors claim that M3-dependent TRPA1 hypersensitivity occurs following CFA but neither the amplitude of TRPA1 responses (seems as if just 2 outliers are driving this difference; Figure 2) nor the percentage of neurons responding to TRPM3+TRPA1 are very supportive of this claim. Furthermore, the increase in A1 sensitivity does not appear to be specific to paw-innervating (i.e. WGA-labeled) neurons, because the same increase (based on a few outliers) occurs in non-paw innervating unlabeled neurons. While the skin imaging data are more convincing, the authors should dampen their claim that TRPM3 expression increases TRPA1 sensitivity following injury.

As outlined in response to reviewer #1, we realize that the original way of representing our data (showing all individual data points and diamond plots) was maybe not the best choice to visualize the magnitude of the effects (although it is a generally recommended way of showing non-parametric data). We have, therefore made important changes to Figure 2, including four figure supplements, to make this aspect clearer for the reader.

The percentage of neurons that show a TRPA1 response is doubled in the retrogradely labeled ipsilateral neurons, and the percentage of neurons that respond both to TRPM3 and to TRPA1 agonism (which include both the M3+A1 (a very small subset) and M3+A1+V1 groups) increases from ~4% to about 16% in this group. This can now be better appreciated from the new Figure 2C and Figure 2—figure supplement 4. Moreover, the average response amplitude to TRPA1 agonists in WGA-AF647^+^ ipsilateral neurons is more than doubled following CFA, which is now also easier to appreciate from the revised Figure 2E.

In addition, we now also provide a value in the text (as well as the corresponding 95% confidence interval [CI]) representing the average increase in response amplitude in ipsilateral WGA-AF647^+^ neurons, which amounts to 229% ([CI], 146% to 355%; P=2×10^-5^) when compared to the WGA-AF647^+^ contralateral neurons. Note that this difference remains highly significant (P=3×10^-5^) if we delete the two highest values, so it is certainly not the case that two outliers drive the difference.

It is true that we also observed increased responses for all three TRP channels in the WGA-AF647^-^ ipsilateral neurons, although (at least for TRPM3 and TRPV1) less pronounced than in the WGA-AF647^+^ neurons. We used a relatively conservative fluorescence threshold for a neuron to be considered WGA-positive. Therefore, there is likely a subset of neurons that innervate the injured paw but contain only a low level of WGA-AF647, and are therefore included in the WGA-negative group. In addition, it cannot be excluded that there is some cross-talk at the level of the DRG between neurons that innervate the inflamed tissue and neighboring DRG neurons. This aspect is briefly discussed in the revised manuscript.

• Since TRPV1-GCaMP3 mice are being used for both DRG and skin calcium imaging, how is it possible that not all cells respond to TRPV1 agonists? In other words, how are M3 only, A1 only, or M3+A1 responders detectable using this genetic line since only TRPV1 positive neurons would express GCaMP3? It seems that this would limit the authors' ability to detect changes in A1 sensitivity following injury. If this population of TRPV1-lineage neurons includes all of the TRPM3 and TRPA1-expressing neurons in the adult, then this should be made even more clear in the manuscript.

This is a misunderstanding, and we apologize for not explaining this issue better. The TRPV1-cre driver line, which was used to generate the TRPV1-GCaMP3 mice, expresses cre in all TRPV1-lineage neurons, i.e., all neurons that express TRPV1 at any time in their development. In the studies of Mishra et al. (Mishra and Hoon, 2010 and Mishra et al., 2011), it was shown that all sensory neurons involved in thermosensation inflammatory heat hyperalgesia express TRPV1 at an early developmental stage, but that a significant proportion (roughly 50%) of these TRPV1-lineage neurons lose expression of TRPV1 in adult mice. Since these neurons have expressed the cre recombinase in an early stage, they express GCaMP3 even though they are no longer TRPV1 positive. This population includes the M3 only, A1 only or M3+A1 responders, as well as TRPM8-responding cells, as nicely demonstrated in Mishra et al., 2011.

To make this point clearer, we have now included an analysis of combined GCaMP3 and Fura-2 measurements of isolated neurons from the TRPV1-GCaMP3 mice as Figure 2—figure supplement 1. These results show that ~60% of the GCaMP3-positive and 0% of the GCaMP3-negative neurons respond to capsaicin. The ~40% GCaMP3-positive neurons that do not respond to capsaicin are those neurons that expressed TRPV1 during development but lost expression at the stage where we did our analysis. These indeed include neurons that respond to TRPM3 and TRPA1 agonism. In addition, we found that ~40% of the GCaMP3-positive and 0% of the GCaMP3-negative neurons respond to MO. This indicates that all TRPA1-positive neurons are indeed included within the analyzed set of neurons. This finding is also fully in line with the earlier findings showing that the ablation of TRPV1-lineage neurons fully abolished the expression of TRPA1 (Zariwala et al., 2012). Finally, we found that about 60% of the GCaMP3-positive but also ~10% of the GCaMP3-negative neurons respond to PS. This suggests that most, but not all, TRPM3-expressing DRG neurons are within the set of TRPV1-lineage neurons. This is also in line with our RNAscope-data, showing a higher percentage of TRPM3-positive DRG neurons than of TRPV1-positive DRG neurons.

• The authors should discuss whether there is potential cross desensitization/sensitization between agonist in both the DRG and the peripheral skin calcium imaging experiments, since cross-interactions between agonists could limit the strength of their conclusions.

In preliminary experiments, as well as in earlier work in isolated sensory neurons, we observed that capsaicin treatment leads to significant desensitization of sensory neurons to subsequent capsaicin/AITC/PS stimulation, whereas this is much less pronounced for AITC and not observed for PS. We, therefore, chose to use the stimulation order PS/ AITC/ capsaicin in all experiments, as also done in our earlier work (Vandewauw et al., 2018; Vriens et al., 2011). We have included a brief discussion of the issue of cross (de)sensitization in the revised manuscript. Whereas the full intricacies of the (potential) direct or indirect interactions between the three channels remain to be uncovered, we don't think such interactions would significantly limit the strength of our conclusions. The primary novel finding, namely the increased activity of TRPM3, is certainly not compromised, as this channel was always probed first. Moreover, we believe that by using an identical stimulation protocol on all preparations, a direct comparison between the groups is certainly valid.

• More information/citations should be provided as to the dose/concentration chosen for the TRPM3 antagonist isosakuranetin, particularly since a single concentration was used of this single antagonist.

Additional information is included in the revised manuscript. We now refer to papers that show that the concentration of 20 µM is in line with free plasma levels after systemic application of isosakuranetin in mice at a dose that inhibits TRPM3 activity in vivo.

• The authors are commended for ex vivo imaging of sensory afferent terminals in the skin. Depending on the scientific question, this approach could have many benefits over imaging exclusively at the level of the DRG. Since this is the first description of this method, the authors should further clarify the methods in detail and address the following questions:How large is the bath containing the skin? How does buffer exit the bath? It appears that agonists are being presented to the tissue for 30-45 s. with an equivalent washout period between compounds (Figure 3C). Is this sufficient time for the agonists to both access sensory terminals deep in the skin and then be washed away from the tissue? More details are needed on whether there is potential desensitization from one agonist to the next if insufficient time is allowed for agonist washout. More details on this method, the regions of interests chosen and how the data are quantified in a blinded manner should be provided.

We included more details in the revised manuscript, including a clear time course of the agonist application periods for the experiments in Figure 4. As is shown there, we allowed 5 minutes in between stimuli.

• The authors should comment on the use of a saphenous (presumably dorsal hairy skin) preparation use in the context of the current injury model (i.e. 10 µL injection of CFA to the plantar surface of the paw)?

The dorsal hairy skin turned out to be much better for imaging than the thicker and more irregular plantar side. We confirmed that CFA injection resulted in an inflammation of the entire paw. More details on these issues are now included in the revised manuscript.

I don't think that the use of the contralateral paw is the best control since there could be segmental effects, and separate animal control would be better. That said, this would require the entire data set to be re-generated and I'm not going to recommend this. But this is something that I think that the authors should note going forward and avoid using the contralateral paw as the control.

There could indeed be alterations on the contralateral side, although, at least behaviorally, effects on the contralateral side have not been observed in the CFA model after 24 h. Using separate animal controls would require at least twice as many animals, and probably many more because paired statistics would no longer be possible. Moreover, by using both paws from the same animal, we compensate for inter-animal variations in skin thickness and innervation. Therefore, for the CFA model, we believe that our approach is adequate and justified in our striving to reduce the use of animals to the minimum. We do, however, fully agree with the reviewer that separate animal controls would be more apt for more chronic pain models (e.g., PSNL neuropathic pain) where clear contralateral sensitization takes place.

[Editors’ note: what follows is the authors’ response to the second round of review.]

The editors and reviewers found the revised manuscript to be greatly improved, and we appreciate your sincere efforts to address the concerns that were raised. Overall, the conclusions better align with the results obtained, although we have some remaining concerns with conclusions concerning the RNAscope and skin prep results.RNA scope:The example images in Figure 1 are improved, however the new staining for Trpm3 looks to either be quite non-specific or to have high background? Either might affect the conclusions. In addition, the reviewers remain unconvinced that TRPM3 expression increases in response to inflammation. Some changes might seem big percent-wise but cell to cell variation and the biological meaning of such differences should also be considered: for example, what's the difference between an average of 20 and 30 puncta when the range of variation between cells is far greater? While these results may be suggestive, our consensus is the RNAscope results are not conclusive. We were much more convince by the increased functional responses to TRPM3 agonists. We request that you tone down conclusions concerning the RNAscope results throughout the manuscript. For example, in the Results section the following sentence:“Taken together, these results indicate that inflammation is associated with a significantly increased transcription of TRPM3, specifically in sensory neurons innervating the inflamed tissue, whereas no inflammation-related changes were found in the mRNA levels of TRPV1 and TRPA1.”could be easily improved by replacing 'indicate' with 'suggest' and "a significantly" with 'an'.

We have toned down the conclusions regarding the RNAscope along the suggested lines.

While we recognize the reviewers’ reservations, we do not think that the Trpm3 staining is non-specific or has high background. Compared to TRPA1 and TRPV1, TRPM3 is expressed in a higher proportion of the neurons, but never at very high levels (mainly <100 dots/cell). For TRPA1 and TRPV1 there are cells with very high expression levels (>300 dots per cell; see Figure 1—figure supplement 2) along with a large proportion of cells without any RNAscope signal. These findings are in line with functional studies in isolated DRG neurons, which show that a larger fraction of sensory neurons respond to TRPM3 agonists than to capsaicin, but also that the amplitude of TRPM3-mediated currents is an order of magnitude smaller than that of capsaicin-activated currents (see e.g. Vriens et al., 2011).

Skin assay:The limitations of the assay are clearly spelled out in the Results section, but as analyzed the results are inherently qualitative. As such we suggest also toning down the conclusion here. For example, in the results the following sentence:“Overall, these results represent, to our knowledge, the first direct observation of functional upregulation of all three heat-activated TRP channels in intact nerve endings in inflamed skin.”could be improved by inserting 'qualitative' before 'results' to remind the reader of the nature of the measurements.

We have toned down the conclusions regarding the skin assay along the suggested lines in several instances.

While we acknowledge (and discuss in the manuscript) that quantification of calcium signals in nerve endings in the skin is not as straightforward as in cell bodies, we do not think that the results obtained with this approach are inherently purely qualitative. Our analysis pipeline provides quantitative values representing the area of nerve terminals responding to agonists, which in our opinion is a valid measure of channel activity in nerve endings.